# V2V: Scaling Event-Based Vision through Efficient Video-to-Voxel Simulation

**Hanyue Lou**[1,2]  **Jinxiu Liang**[3]  **Minggui Teng**[1]  **Yi Wang**[2,4]  **Boxin Shi**[1#]

[1] Peking University  [2] Shanghai Innovation Institute
[3] National Institute of Informatics  [4] Shanghai AI Laboratory
{hylz, minggui_teng, shiboxin}@pku.edu.cn
cssherryliang@gmail.com  wangyi@pjlab.org.cn

## Abstract

Event-based cameras offer unique advantages such as high temporal resolution, high dynamic range, and low power consumption. However, the massive storage requirements and I/O burdens of existing synthetic data generation pipelines and the scarcity of real data prevent event-based training datasets from scaling up, limiting the development and generalization capabilities of event vision models. To address this challenge, we introduce Video-to-Voxel (V2V), an approach that directly converts conventional video frames into event-based voxel grid representations, bypassing the storage-intensive event stream generation entirely. V2V enables a 150× reduction in storage requirements while supporting on-the-fly parameter randomization for enhanced model robustness. Leveraging this efficiency, we train several video reconstruction and optical flow estimation model architectures on 10,000 diverse videos totaling 52 hours—an order of magnitude larger than existing event datasets, yielding substantial improvements.

## 1 Introduction

Event-based cameras [8] are bio-inspired visual sensors that asynchronously record per-pixel intensity changes, offering high temporal resolution and high dynamic range with low power consumption, compared to traditional frame-based cameras. These properties make event cameras particularly promising for applications involving high-speed motion, challenging lighting conditions, and resource-constrained environments such as robotics, autonomous driving, and augmented reality.

Despite their theoretical advantages, event cameras have yet to match the practical performance of conventional cameras. This performance gap can be attributed to several factors, with the scarcity of large-scale training data constituting a critical bottleneck. While deep learning approaches have revolutionized frame-based computer vision through decades of dataset collection and massive repositories like ImageNet [4], COCO [19], and WebVid [2], event camera data remains scarce and limited in diversity. The restricted commercial deployment of event cameras creates a fundamental chicken-and-egg problem: widespread adoption requires robust algorithms, which in turn depend on diverse training data that can only come from widespread deployment.

To address this data limitation, researchers have developed simulation pipelines that convert conventional visuals into synthetic event ones. However, significant challenges remain in both approximating the high temporal dynamics of event cameras and utilizing these simulations efficiently.

Standard video datasets typically have frame rates of around 30 per second (FPS), providing temporal resolution orders of magnitude lower than event cameras. This fundamental mismatch creates a

---

[1] Hanyue Lou, Minggui Teng and Boxin Shi are with the State Key Laboratory of Multimedia Information Processing and the National Engineering Research Center of Visual Technology, School of Computer Science, Peking University.  [#] Corresponding author.  [$] Code available: `https://github.com/HYLZ-2019/V2V`

39th Conference on Neural Information Processing Systems (NeurIPS 2025).

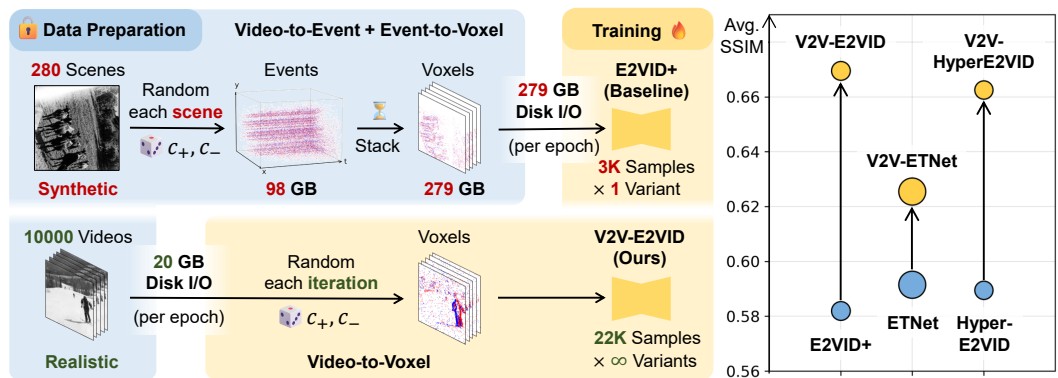

Figure 1: The Video-to-Voxel (V2V) approach enables our training dataset to have more diversity (280→10000 scenes), less storage usage and I/O load (279 GB→20 GB), and more randomly augmented sample variants. V2V-retrained models E2VID [33], ETNet [36] and HyperE2VID [7] demonstrate better reconstruction quality, indicated by higher SSIM (structural similarity, averaged over HQF [33] and EVAID [6]) on real-world test datasets.

significant hurdle for data simulation. The field has consequently developed two primary simulation approaches, each with distinct limitations: **Video-based simulators** like V2E [13] attempt to bridge this temporal gap through frame interpolation techniques; however, they inevitably introduce artifacts and interpolation errors since they attempt to reconstruct information that was never captured in the original footage. **Model-based simulators** like those in E2VID+ [33] render high frame rate videos from 3D scene models, providing precise control of the temporal resolution but suffering from significant realism limitations. In practice, these methods often resort to simplistic approximations, such as randomly flying 2D images against static backgrounds – far removed from real-world scene complexity, as illustrated in topleft of Figure 1.

Beyond realism concerns, both approaches encounter severe storage and computational inefficiency. Unlike conventional video data, which benefits from decades of codec optimization, event data compression remains relatively underdeveloped [14, 31]. Consequently, storing and processing synthetic event data becomes exceedingly resource-intensive. For context, a single 10-second sequence from the ESIM-280 dataset occupies approximately 1GB. If we were to scale to 10,000 diverse scenes, substantial I/O bottlenecks during training will emerge. Moreover, since each event stream corresponds to a specific camera configuration, exploring multiple parameter variations (thresholds, noise levels, *etc*.) multiplies storage requirements by the number of configurations.

These technical challenges create a significant barrier to training on large-scale, diverse datasets, leading us to question: *Is explicit event stream synthesis truly necessary for model training?*

Our analysis reveals a surprising answer: *No*. We observe that most event-based deep learning pipelines convert asynchronous events into dense representations (particularly voxel grids) before neural network processing. This creates an opportunity to directly target these representations instead of generating intermediate event streams.

In this paper, we introduce Video-to-Voxel (V2V), a principled approximation that efficiently generates discrete voxel representations directly from video frames. This approach bypasses the computationally expensive event generation step while preserving the essential spatial-temporal information needed for learning. The key advantages of our approach are illustrated in Figure 1:

- Efficiency. Our approach eliminates the need to store intermediate event streams, drastically saving storage resources (up to 150 times) and enabling the use of substantially larger datasets.
- Flexibility. Our approach allows on-the-fly parameter randomization during training, enabling models to learn more robust representations from diverse virtual camera configurations without multiplicative storage requirements.
- Scalability. This efficiency enables direct utilization of massive internet-scale video repositories like WebVid [2], creating a training dataset containing 10,000 videos with a total duration of 52 hours—an order of magnitude larger than existing event camera datasets.

We validate the effectiveness of the V2V module on two fundamental event-based visual dense prediction tasks: event-based video reconstruction and optical flow estimation. In particular, our method enables the training of existing models on a diverse dataset with 35 times more scene variations than the previous practice, with performance scaling with dataset size. As shown in the right of Figure 1, this leads to significant improvements in reconstruction quality. These results demonstrate that our V2V pipeline effectively addresses the data scarcity bottleneck in event-based vision, opening new possibilities for developing robust event-based algorithms.

## 2 Related Works

**Synthetic event data.**    Video-based and model-based event simulators have been proposed to generate synthetic event data. Video-based simulators [13, 18, 41, 44, 46] synthesize events from videos, using frame interpolation algorithms [13] or learning-based methods [41, 44] to compensate the lack of temporal information due to limited video frame rate. This requries reconstructing intermediate intensity changes that were never captured—an ill-posed problem. Model-based event simulators such as ESIM [28] and PECS [12] render events from synthetic 3D scenes, providing fully precise temporal information. However, the diversity and realism of the scenes are limited. Random scenes with flying 2D [33] or 3D [35] objects and elaborate models of city scenes [22] have been used, but complex real-world phenomena, such as with non-rigid movement, fluid dynamics, and complex lighting interactions, remain difficult to model.

**Training data of event-based video reconstruction.**    The development of video reconstruction from event streams (E2VID) has been predominantly constrained by available training data. Existing learning-based E2VID methods use model-based simulated events or real events for training. E2VID [29] proposed to use the ESIM simulator for training data generation, and Stoffregen *et al*. [33] improved model performance by using diversified thresholds in ESIM. FireNet [30], SPADE-E2VID [3], ETNet [36], Event-Diffusion [17], HyperE2VID [7], EVSNN [45] and TFC-SNN [38] all used ESIM data for training. Mostafavi *et al*. [24] used a mixture of ESIM data and real data. Gu *et al*. [11] used the video-based simulator V2E [13], but the high frame rate video inputs were from synthesized scenes with flying images. SSL-E2VID [27] proposed a self-supervised training framework so that the model could be trained on real event streams without corresponding ground truth images. The low light methods DVS-Dark [40], NER-Net [20], and NER-Net+ [21] utilize real-world low-light events for domain adaptation.

**Training data of event-based optical flow estimation.**    Event-based optical flow estimation has explored marginally more diverse training strategies but encounters similar limitations. EVFlow [42] is trained on ESIM-generated data, while EVFlow+ [33] is trained on ESIM with diversified thresholds. ADMFlow  [22] utilizes a synthetic dataset MDR, rendered by browsing cameras through 53 static virtual 3D scenes and using the V2E [13] toolbox. E-FlowFormer [16] is trained on BlinkFlow, which is composed of 3362 rendered scenes of random moving objects. Zhu *et al*.trained their model on the real MVSEC dataset [43] with an unsupervised loss, while Spike-FlowNet [15] and STE-FlowNet [5] used it in a self-supervised way. ERAFT [10] is trained on DSEC-Flow, a real dataset from 24 DSEC [9] sequences. EEMFlow+ [23] is trained on HREM, a real event dataset with 100 scenes.

## 3 Method

In this section, we will introduce our V2V framework, which utilizes large-scale video datasets for training. In Section 3.1, we introduce preliminaries and emphasize the importance of data scalability. In Section 3.2, we reveal that intra-bin temporal information can be neglected in training with the discrete voxel representation, greatly improving efficiency. Finally, in Section 3.3, we show that on-the-fly video-to-voxel generation provides additional flexibility that improves data diversity.

### 3.1 Preliminaries

**Formulation of event generation model.**    Event cameras are bio-inspired vision sensors that operate on fundamentally different principles than conventional frame-based cameras. Instead of capturing intensity frames at fixed time intervals, event cameras asynchronously report changes in logarithmic brightness at the pixel level. In an ideal event camera, a pixel at $(x, y)$ triggers an event

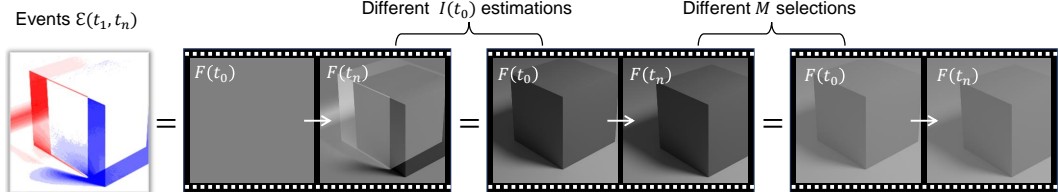

Figure 2: Event data's inherent ambiguity: multiple valid image sequences $I(t_0)$ can produce identical event streams $\mathcal{E}$ under different initial conditions and camera parameters $M$. It indicates the critical importance of diverse training data to establish robust priors for reconstruction tasks.

signal $(x, y, t, p)$ whenever the change of its logarithmic irradiance exceeds a threshold $c$:

$$|\log I_{x,y}(t) - \log I_{x,y}(t_0)| \geq c, \tag{1}$$

where $I_{x,y}(t)$ indicates the pixel's irradiance at time $t$, and $t_0$ is the timestamp of the previous event triggered from pixel $(x, y)$. The polarity $p \in \{-1, +1\}$ represents whether the irradiance has decreased or increased.

The output of an event camera is a stream of asynchronous events $\mathcal{E}$, capturing the *change* of the scene. By summing the positive and negative events triggered between $t_1$ and $t_2$ on each pixel into an event stack $E(t_1, t_2)$, we can acquire the relationship between irradiance levels $I(t_1)$ and $I(t_2)$:

$$\log I(t_2) - \log I(t_1) = c * E(t_1, t_2), \quad \text{where} \quad E_{x,y}(t_1, t_2) = \sum_{(x,y,t,p) \in \mathcal{E}, t_1 \leq t < t_2} p. \tag{2}$$

**The inherent ambiguity in event-based vision tasks.** This relationship highlights a fundamental challenge in event-based vision which has been widely discussed [29, 34]: events only provide constraints on relative changes in logarithmic irradiance, not absolute intensity values. For example, in the task of event-to-video reconstruction, a model is trained to reconstruct a series of image frames $F(t_1), \ldots, F(t_n)$ based on an event stream $\mathcal{E}$. This means that it must estimate both an initial irradiance $I(t_0)$ and a mapping $M$ from irradiance $I(t_i)$ to pixel values $F(t_i)$ that would produce high-quality frames, which are related to camera parameters such as aperture size, exposure time and image signal processer (ISP) configurations. As demonstrated in Figure 2, multiple valid interpretations of the same event stream exist, corresponding to different scene priors and camera parameters. To provide models with high-quality prior knowledge, large-scale datasets that resemble real-world distributions are essential.

However, current model-based simulators cannot achieve scale, diversity, and quality simultaneously, as realistic 3D models are prohibitively expensive to design and simple scenes fall short of real-world complexity. Utilizing existing video datasets offers a promising alternative, but requires addressing the significant temporal resolution gap between conventional videos (typically 30 FPS) and event data (microsecond resolution). Our method bridges this gap through a novel representation approach that enables effective learning from standard video data while preserving compatibility with real event camera outputs.

## 3.2 Bridging the temporal resolution gap

The fundamental challenge in utilizing conventional videos for event-based learning lies in the vast difference in temporal resolution. Event streams operate at microsecond temporal precision, while typical videos provide only 30-60 frames per second. Although frame interpolation algorithms could theoretically increase frame rates, interpolating to event-level frame rates would be computationally prohibitive and introduce compounding artifacts such as ghosting effects [37]. More importantly, such interpolation attempts to reconstruct temporal information that was never captured in the original footage, resulting in significant fidelity loss.

Our approach tackles this challenge by fundamentally reconsidering how event data is represented and processed in deep learning pipelines. For deep learning approaches, the sparse, asynchronous nature of event streams necessitates conversion into dense representations that can be fed into conventional neural networks. Among these representations, voxel grids have emerged as the most widely used format. In voxel representations, events are first separated into temporal bins, and each bin of events

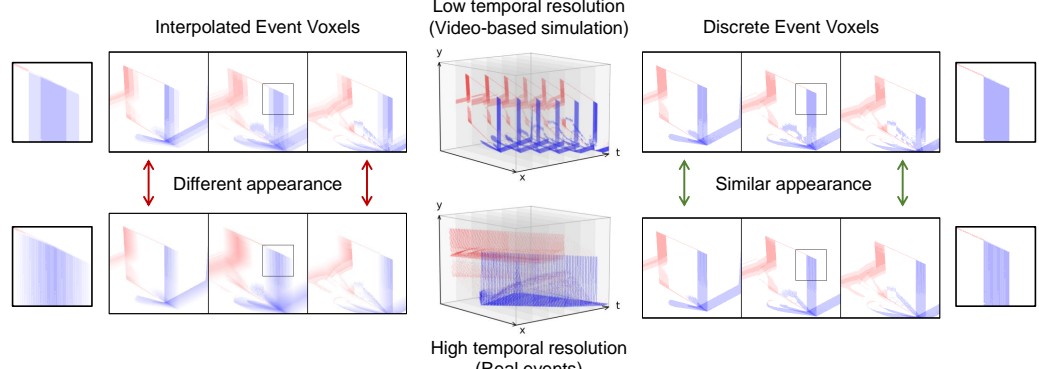

Figure 3: While synthetic (top) and real (bottom) events exhibit notable differences in interpolated voxels (left) due to microsecond-level temporal disparities, they demonstrate similarity in discrete voxel representations (right)—justifying our direct video-to-voxel conversion approach.

is encoded into a frame through some form of accumulation. These frames are then stacked into different channels of the voxel corresponding to their sequence. This organization creates a natural separation between *inter-bin* temporal information (the relationship between separate bins) and *intra-bin* temporal information (the precise timing of events within each bin). This separation is crucial to our approach, as it allows us to discard the high-precision intra-bin timing information—which cannot be reliably simulated from low frame rate videos—while preserving the essential inter-bin dynamics that can be accurately derived from frame differences.

**Interpolated *vs.* discrete voxel grids.** The most commonly used voxel representation is the interpolated event voxel. An interpolated event voxel $V_{\text{intp}}$ with $B$ bins has shape $B \times H \times W$. For each event $(t, x, y, p) \in \mathcal{E}$, its polarity is linearly split to the closest two bins according to its timestamp $t$ (normalized to $[0, 1]$):

$$V_{\text{intp}}(b, x, y) = \sum_{(x,y,t,p) \in \mathcal{E}} p \, \max(0, 1 - |(B-1)t - b|). \tag{3}$$

The interpolated event voxel representation is widely used in event-based video reconstruction algorithms [3, 17, 27, 29, 33, 36] and optical flow estimation algorithms [10, 22, 33]. However, it does not meet our needs. The intra-bin temporal information still takes significant effect: as the timstamp of an event changes, its weights on the two neighbouring bins continuously changes, resulting in smooth smudge-like edges in voxels. When the intra-bin temporal information is absent, the discreteness of the timestamps creates layered-like visual effects in the voxels. The difference is depicted in the left wing of Figure 3.

The discrete event voxel, however, meets our needs. It is another variant of the voxel representation, also referred to as SBT [24]. Each bin in a $B \times H \times W$ discrete event voxel $V_{\text{disc}}$ is the summation of all events with timestamps between $b/B$ and $(b+1)/B$:

$$V_{\text{disc}}(b, x, y) = \sum_{(x,y,t,p) \in \mathcal{E}} p \, \mathbf{1}[\frac{b}{B} \leq t < \frac{b+1}{B}]. \tag{4}$$

In the discrete event voxel, all intra-bin information is discarded, and temporal information is only encoded in inter-bin information. As shown in the right side of Figure 3, the discrete voxel representations of video-based simulated events show similar appearances to real ones with high temporal resolution. One may worry that discarding intra-bin information would hurt the expressiveness of the input data, but our experimental results ((l)&(m) in Table 2) show that models operating on the interpolated and discrete representations can achieve comparable performance.

For a discrete event voxel with $B$ bins, we can efficiently simulate it with just $B + 1$ video frames $F(t_0), F(t_1), \cdots F(t_B)$, as detailed in the next section. If the input video has a frame rate $X$ FPS, then each created voxel will cover physical time of $B/X$ seconds. A potential concern is that the physical time span of real event voxels may be much shorter when we want to exploit its high temporal resolution. However, due to the large scale and high diversity of our video datasets, we

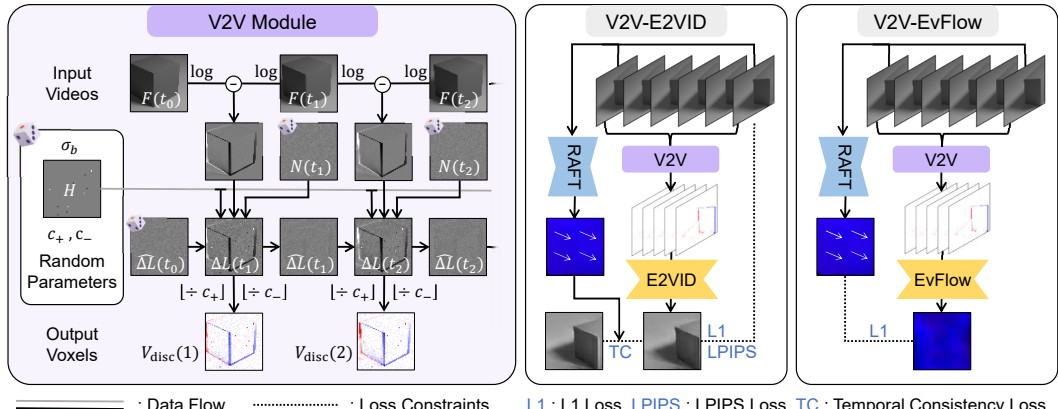

Figure 4: The V2V module efficiently converts input videos to output voxels with random parameters selected at train time (left). We use the V2V module and a lightweight optical flow estimator RAFT to train video reconstruction (middle) and optical flow estimation models (right).

find that our models generalize well: although trained on $24/5$ FPS synthetic voxels, they achieve superior performance in reconstructing 150 FPS videos from real EVAID events [6].

This representation choice is the cornerstone of our approach, enabling us to significantly reduce storage requirements and improve training efficiency by leveraging advanced video compression and decoding techniques.

## 3.3 Directly converting videos to voxels

There exist many simulation algorithms that convert videos to events, such as EventGAN [44], V2E [13], DVS-Voltmeter [18] and V2CE [41]. They are usually used in a two-stage pipeline. In the first stage, event camera parameters such as thresholds and noise strength are selected. The photoreceptor output voltage of the event sensor is simulated using the logarithm difference between video frames as well as random noise. The number of events to be triggered on each pixel, equivalent to the values of the corresponding discrete voxel, is calculated according to the thresholds. Event streams are then generated by deciding a timestamp for each event triggered. In the second stage, events are accumulated into dense representations such as voxels. Additional augmentation such as hot pixels and gaussian noise can be added to the voxels on-the-fly when training, but with less fidelity. For example, in real event cameras, when background noise triggers an event, the brightness change detector is reset, effecting future pixel behaviour, while post-voxel augmentation noise is independent to the signals.

Our Video-to-Voxel (V2V) approach skips the generation of event streams, merging both stages into an efficient on-the-fly process conducted in each training iteration. As a result, the camera parameters and high-fidelity noise corresponding to a single video can be randomly selected in each iteration, providing far more sample diversity and improving model robustness. The V2V conversion process is conducted in the following steps.

**Parameter selection.** In each V2V iteration, we first randomly select the event camera parameters: the positive threshold $c_+ > 0$, the negative threshold $c_- > 0$, background noise strength $\sigma_N > 0$ and a hot pixel map $H \in \mathcal{R}^{H \times W}$. The background noise corresponds to the dark current in the photodiode of event pixels, and the hot pixel map $H$ is a sparse matrix where a small number of randomly selected "hot pixels" have abnormally large values and the other elements are zero [13].

**Initialization.** Then we begin the simulation. We use $\Delta L$ to denote the sensor's photoreceptor output voltage, corresponding to the amount of logarithm illumination change on each pixel since their last previous triggered event. The initial value $\Delta \hat{L}(t_0)$ is initialized by sampling from the uniform distribution $[-c_-, c_+]$, assuming that the "sensor" has already been recording for a while.

**Sensor simulation.** We process the frames $F(t_0), F(t_1), ..., F(t_B)$ one by one. Given the frame $F(t_i)$, we first apply a reverse gamma correction to make it approximately linear to $I(t_i)$, and calculate its logarithm difference to the previous frame $\log I(t_i) - \log I(t_{i-1})$. Following DVS-

Voltmeter [18], which models sensor voltage change as a Brownian motion process caused by photon reception randomness, we add a background noise independently sampled from a gaussian distribution $\mathcal{N}(0, \sigma_b^2)$ for each frame. We also randomly generate a hot pixel map $H$ that keeps the same throughout the sequence. For simplicity, we did not include other noise types like leak noise, but they could be easily incorporated into our framework. We add up the signal and noise to acquire the total output voltage:

$$\Delta L(t_i) = \Delta \hat{L}(t_{i-1}) + (\log I(t_i) - \log I(t_{i-1})) + N(t_i) + H, \quad N(t_i) \sim \mathcal{N}(0, \sigma_b^2). \quad (5)$$

Then, we calculate the number of positive and negative events triggered, denoted as $N_+$ and $N_-$, and subtract these already-triggered changes to prepare for the next input frame:

$$N_+(i) = \max \left( 0, \left\lfloor \frac{\Delta L(t_i)}{c_+} \right\rfloor \right), \quad N_-(i) = \max \left( 0, \left\lfloor \frac{-\Delta L(t_i)}{c_-} \right\rfloor \right), \quad (6)$$

$$\Delta \hat{L}(t_i) = \Delta L(t_i) - c_+ N_+(i) + c_- N_-(i). \quad (7)$$

Note that, unlike post-voxel augmentation noise, all noise added in the V2V process can play an interactive part in the event simulation, which has the potential to lead to higher data fidelity.

**Voxel calculation.** Finally, we can compute the corresponding discrete event voxel bin:

$$V_{\text{disc}}(i) \approx N_+(i) - N_-(i). \quad (8)$$

When $c_+$ and $c_-$ are equal, or when the pixel intensity change between two frames is monotonic, the equality holds strictly.

**Applications.** As illustrated in Figure 4, we apply the V2V module to the video reconstruction (E2VID) and optical flow estimation (EvFlow) tasks by using them to generate voxels from video-based datasets. We predict optical flow used for training from the video frames, using image-based algorithms such as RAFT [32]. The V2V module enables us to train efficiently on large-scale diverse datasets with flexible augmentation, boosting the performance of the models it is applied to.

## 4   Experiments

| Dataset | Scenes | Duration | Resolution | Seqs | Space |
|---------|--------|----------|------------|------|-------|
| ESIM-Event | 280 | 47 min | $256 \times 256$ | 3421 | 97.77 GB |
| ESIM-Voxel | 280 | 47 min | $256 \times 256$ | 3421 | 279.00 GB |
| WebVid100 | 100 | 33 min | $180 \times 596$ | 253 | 0.21 GB |
| WebVid1k | 1000 | 319 min | $180 \times 596$ | 2332 | 1.94 GB |
| WebVid10k | 10000 | 52 hours | $180 \times 596$ | 22725 | 19.14 GB |
| IJRR [25] | 27 | 18 min | $240 \times 180$ | 638 | 32.2 GB |
| MVSEC [43] | 14 | 68 min | $346 \times 260$ | 9063 | 109.30 GB |
| HQF [33] | 14 | 11 min | $240 \times 180$ | 385 | 2.74 GB |
| EVAID-R [6] | 14 | 6 min | $954 \times 636$ | 1350 | 7.4 GB |

Table 1: Comparison of dataset characteristics. "Seqs" for event datasets represents total frames divided by 40, providing a normalized comparison metric.

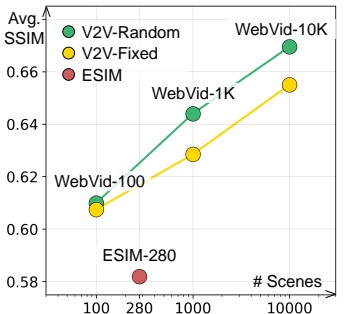

Figure 5: Effectiveness of the proposed parameter randomization across dataset sizes.

### 4.1   Dataset preperation

We use the WebVid [2] video dataset for model training due to its high aesthetic quality and clean shot cuts. From the 2.5M videos of WebVid, we randomly sample 10K, 1K and 100 videos, forming our datasets WebVid10K, WebVid1K and WebVid100. In Table 2, we use "V2V-X" to represent our V2V framework combined with the WebVid-X dataset.

The original resolution of WebVid videos is $336 \times 596$, and we only use the upper $180 \times 596$ to avoid the watermarks. For each WebVid training sequence, we use 200 non-overlapping frames to create 40 voxels, each with 5 event bins. The average WebVid video contributes 2 samples.

We prepare the ESIM-280 dataset using the same configurations as Stoffregen *et al.* [33], generating 280 scenes composed of 2D flying objects along with camera movement. We also sample non-overlapping sequences with 40 event voxels and ground truth frames, acquiring 3421 sequences ("Seqs" in Table 1) with $256 \times 256$ resolution in total. We refer to the original dataset as ESIM-Events, and the pre-stacked dataset with voxels (in float32 format) instead of events is referred to as ESIM-Voxel.

The statistics of our WebVid datasets and the ESIM datasets are listed in Table 1. Due to the compact encoding of videos, WebVid10K only uses $6.9\%$ disk space while providing $6.6\times$ more sequences and $35.7\times$ more scene diversity compared to ESIM-Voxel. It also provides rich real-world priors with its realistic scenes, diverse viewing angles and non-rigid movement such as human motion.

Table 2: Quantitative evaluation of event-based video reconstruction results. Lower (↓) MSE and LPIPS values and higher (↑) SSIM values are desirable. For each model type, we highlight the best values with green . Experiments marked with ⊥ use the original model weights directly downloaded.

| Idx | Model | Train Dataset | Loss | Eps | HQF | | | EVAID | | |
|-----|-------|---------------|------|-----|------|-------|--------|------|-------|--------|
| | | | | | MSE↓ | SSIM↑ | LPIPS↓ | MSE↓ | SSIM↑ | LPIPS↓ |
| *Results from different training dataset sizes* | | | | | | | | | | |
| (a) | E2VID | ESIM-280 ⊥ | Alex+TC | 500 | 0.037 | 0.638 | 0.256 | 0.088 | 0.526 | 0.469 |
| (b) | E2VID | V2V-100 | V+L1+H | 8000 | 0.042 | 0.626 | 0.327 | 0.078 | 0.594 | 0.441 |
| (c) | E2VID | V2V-1K | V+L1+H | 800 | 0.037 | 0.661 | 0.286 | 0.072 | 0.627 | 0.405 |
| (d) | E2VID | V2V-10K | V+L1+H | 80 | 0.032 | 0.676 | 0.265 | 0.054 | 0.663 | 0.371 |
| *Results with other model structures* | | | | | | | | | | |
| (e) | HyperE2VID | ESIM-280 ⊥ | Alex+TC | 400 | 0.032 | 0.646 | 0.265 | 0.071 | 0.533 | 0.474 |
| (f) | HyperE2VID | V2V-10K | V+L1+H | 60 | 0.035 | 0.671 | 0.269 | 0.055 | 0.654 | 0.377 |
| (g) | ETNet | ESIM-280 ⊥ | Alex+TC | 700 | 0.035 | 0.642 | 0.274 | 0.080 | 0.541 | 0.523 |
| (h) | ETNet | V2V-10K | V+L1+H | 100 | 0.039 | 0.641 | 0.306 | 0.056 | 0.610 | 0.406 |
| *Ablation study: Effect of augmentation flexibility* | | | | | | | | | | |
| (i) | E2VID | V2V-100-F | V+L1+H | 8000 | 0.049 | 0.611 | 0.330 | 0.073 | 0.604 | 0.451 |
| (j) | E2VID | V2V-1K-F | V+L1+H | 800 | 0.039 | 0.640 | 0.293 | 0.066 | 0.617 | 0.413 |
| (k) | E2VID | V2V-10K-F | V+L1+H | 80 | 0.032 | 0.666 | 0.264 | 0.058 | 0.644 | 0.378 |
| *Ablation study: Interpolated voxel vs. Discrete voxel* | | | | | | | | | | |
| (l) | E2VID(Intp.) | ESIM-280 | Alex+TC | 500 | 0.041 | 0.599 | 0.306 | 0.078 | 0.526 | 0.478 |
| (m) | E2VID(Disc.) | ESIM-280 | Alex+TC | 500 | 0.040 | 0.608 | 0.296 | 0.062 | 0.548 | 0.450 |

## 4.2 Video reconstruction

We retrained the models E2VID [29], ETNet [36] and HyperE2VID [7] on WebVid and ESIM datasets. Then we performed zero-shot model evaluation on the real event datasets HQF [33] and EVAID [6], with metrics MSE (Mean Square Error), SSIM (Structural Similarity) and LPIPS (Learned Perceptual Image Patch Similarity). Quantitative results (a)-(h) in Table 2 and qualitative comparisons (Figure 6) show that models trained with the V2V-WebVid10K dataset can outperform the original ESIM versions.

**Effect of scaling up.** To test the effect of scaling up the dataset, we retrained the E2VID model with datasets of different sizes: WebVid-100, WebVid-1K and WebVid-10K. As shown in rows (b)-(d) in Table 2 and Figure 5, larger datasets produce models with better performance.

**Effect of augmentation flexibility.** A key advantage of our V2V module is that the same $N$ scenes can be augmented with different threshold configurations in each iteration, yielding $N \times M$ sample variants over $M$ training epochs. To validate the importance of this feature by ablation, we designed an experiment where each video in the WebVid dataset was pre-assigned a fixed threshold value, meaning that the model could only access $N \times 1$ sample variants across M epochs. In rows (i)-(k) of Table 2, we denote these processed datasets with "-F" (Fixed). The results are visualized in Figure 5

and show that disabling the augmentation flexibility causes performance drop, proving that providing different variants in each iteration acts as an effective data augmentation technique.

**Perceptual loss design.**    The loss functions of the baseline E2VID+ [33] is composed of a LPIPS loss (from the Alex model) and a temporal consistency loss (from ground truth optical flow). We refer to this combination as "Alex+TC". We observe that using the Alex LPIPS loss causes patterned artifacts in the reconstructed videos, while using a VGG version of the LPIPS loss does not. Since VGG is more inclined to supervise high-level features [39], we combine an L1 loss with the VGG LPIPS loss to supplement the supervision of low-level features. Ablation studies can be found in the appendix (Section D).

**Temporal consistency loss design.**    Due to the lack of ground truth optical flow from videos, we use the RAFT-Small [32] model to predict optical flow between frames at train time. We observe that adding the temporal consistency loss eliminates flickering in the generated videos, but also introduces dirty-window artefacts: static undesired patterns float on top of the videos, as if filming through a dirty window, likely because the model tries to maintain consistency even for unwanted artifacts from previous frames. To mitigate this, we only apply temporal consistency loss to the latter half (last 20 frames of a 40-frame sequence), so consistency is only enforced after the reconstruction has had time to stabilize. In Table 2, we refer to the loss combination of "VGG + L1 + Temporal-Consistency-Half" as "V+L1+H".

**Ablation on voxel representation.**    To show that using discrete voxels does not harm performance, we retrained E2VID with the same loss and the same ESIM-280 dataset, but with the interpolated event voxel (Intp.) and the discrete event voxel (Disc.) as input representations respectively. Results (l)-(m) in Table 2 show that discrete voxels work comparably to interpolated voxels.

**Training details.**    We train the E2VID model with batch size 12, crop size $128 \times 128$ and constant learning rate 0.0001. For ETNet and HyperE2VID, we also follow their original training protocols. The training epochs ("Eps" in Table 2) are set to keep the total amount of iterations approximately the same across experiments.

**V2V parameters.**    For the event threshold parameters $c_+$ and $c_-$ used in the V2V simulator, we first uniformly sample a threshold $c$ from the range $[0.05, 2]$. (Following the ESIM simulator, we calculate logarithm images with `np.log(0.001 + video/255.0)`, so the amount of change between two frames is in the range of $[0, 6.908]$.) To keep the positive and negative thresholds close, we uniformly sample a "threshold ratio" $r$ from the range $[1, 1.5]$, assigning $(c, rc)$ to $(c_+, c_-)$ or $(c_-, c_+)$ with equal probability. The standard deviation of the background gaussian distribution is uniformly sampled from the range $[0, 0.1]$. The standard deviation of the hot pixel gaussian distribution is uniformly sampled from the range $[0, 10]$. The fraction of hot pixels is uniformly sampled from the range $[0, 0.001]$ (i.e., 0 to 0.1%).

More evaluation details (Section A), test results on IJRR [25] and MVSEC [43] (Section E) and more ablation studies (Section D) are provided in the appendix.

### 4.3   Optical flow estimation

To demonstrate that the V2V framework can adapt to various tasks, we also applied it to event-based optical flow estimation. Based on the model design of EvFlow [42], we trained it with the WebVid10K dataset, using optical flow predicted with RAFT-Large from images as pseudo ground truth. We evaluated the models quantitatively on the MVSEC dataset (see Section A for details) with the dt = 1 [42] setting. For metrics we used the Average Endpoint Error (AEE) and the percentage of pixels with more than 3 Pixel Error (3PE). We provide dense (D) metrics which were averaged over all pixels, and sparse (S) metrics which were calculated over pixels that triggered at least one event. We also show qualitative results of our model on IJRR, HQF and EVAID in Figure 17.

In Table 3, Zeros refers to the baseline metrics acquired by predicting zero optical flow. EvFlow+ [33] is trained on the ESIM+ dataset. Although EvFlow+ produces dense results, it often predicts hollows with zero flow where events are sparse. Boosted with the real world priors from the WebVid10K dataset, V2V-EvFlow is able to fill in the hollows with reasonable flow predictions. V2V-EvFlow exceeds EvFlow+ on all metrics, especially dense metrics.

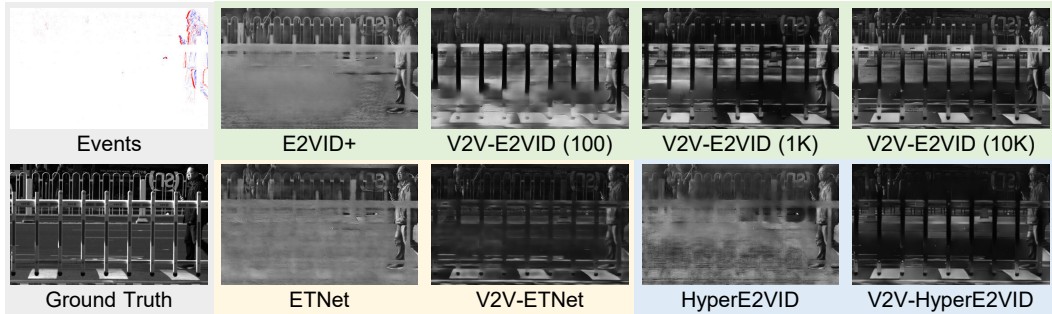

Figure 6: Qualitative comparison of reconstruction results on the EVAID Traffic sequence. Our method maintains information from camera motion occurred previously, even after it ceases.

Table 3: Optical flow results on the MVSEC dataset. Lower (↓) values of AEE and 3PE are desirable, dense (D) or sparse (S). We highlight the best values of each metric with green .

| Sequence | indoor_flying1 | | indoor_flying2 | | indoor_flying3 | | outdoor_day1 | | outdoor_day2 | |
|---|---|---|---|---|---|---|---|---|---|---|
| | D | S | D | S | D | S | D | S | D | S |
| AEE ↓ | | | | | | | | | | |
| Zeros | 1.772 | 2.041 | 2.705 | 3.463 | 2.412 | 2.772 | 3.985 | 3.585 | 2.793 | 2.562 |
| EvFlow+ | 0.967 | 0.752 | 1.154 | 0.990 | 1.184 | 0.833 | 2.903 | 1.445 | 1.983 | 1.580 |
| V2V-EvFlow | 0.732 | 0.745 | 0.870 | 0.990 | 0.741 | 0.758 | 2.114 | 1.191 | 1.627 | 1.365 |
| 3PE (%) ↓ | D | S | D | S | D | S | D | S | D | S |
| Zeros | 14.5 | 20.2 | 36.1 | 52.5 | 30.2 | 38.6 | 57.1 | 54.5 | 32.9 | 31.7 |
| EvFlowt+ | 2.7 | 0.8 | 5.4 | 2.4 | 7.0 | 1.6 | 36.1 | 12.1 | 19.8 | 14.4 |
| V2V-EvFlow | 0.5 | 0.5 | 1.3 | 2.3 | 0.4 | 0.6 | 23.6 | 8.4 | 15.0 | 11.2 |

# 5    Limitations

**Computation resources.**    Since all training data is stored in video format, video decoding is required throughout the training process. Video decoding algorithms are quite CPU-intensive; when accelerated with hardware, they also consume GPUs. In our experiments, the CPU capabilities of our systems were sufficient, and the real efficiency bottleneck was successfully pushed to the 100%-utilized GPU. However, this may be a problem on systems with other CPU-GPU combinations.

**Event representations.**    As analyzed in Section 3, the video-to-voxel module is only applicable when the input representation of a neural network is one that discards all intra-bin temporal information, such as the discrete voxel. Although the discrete voxel representation can be applied to a large range of event-based vision tasks, there are still scenarios when non-applicable representations are essential, especially if the model to train is unconventional (*e.g.*, spiking neural networks).

# 6    Conclusion

We introduce the V2V module, which directly converts videos to voxels and significantly reduces the storage and data transfer costs of synthetic event data while providing more variant training samples. Empowered by the V2V module, we scale up training datasets for video reconstruction and optical flow estimation, boosting existing models to exhibit better performance. The V2V module has broad application potential across more model architectures and event-based tasks, which remains to be explored in the future.

# Acknowledgments

This work was supported by National Natural Science Foundation of China (Grant No. 62088102, 62302019, 62136001), Beijing Natural Science Foundation (Grant No. L233024), and Beijing Municipal Science & Technology Commission, Administrative Commission of Zhongguancun Science Park (Grant No. Z241100003524012).

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

# Appendix

## A   Evaluation Details

In Table 2, we report metrics MSE, SSIM, and LPIPS. There are several versions of these metrics that produce different values, causing inconsistency across papers. We use a version that closely reproduces the values in previous works such as E2VID++ [33] and ETNet [36]. Specifically:

**MSE.**   In our experiments, Mean Square Error is calculated on pixel values of images normalized in the range of $[0, 1]$.

**SSIM.**   Structural Similarity is calculated with `skimage.metrics.structural_similarity` from the Python Package "scikit-image". The images to be processed are normalized into the range of $[0, 1]$. The window size is set to the default `win_size=7`. The `data_range` parameter is incorrectly set to 2 instead of 1 to align with previous works.

**LPIPS.**   The Learned Perceptual Image Patch Similarity is calculated with the code and weights provided in `https://github.com/cedric-scheerlinck/PerceptualSimilarity`. The parameters are set to `net="alex"` and `version="0.1"`. It is noted that using different versions of code and weights will lead to different metric values.

We evaluate the methods on selected sequence cuts of IJRR [25], MVSEC [43], HQF [33], and EVAID [6]. Following [33], we use the full HQF sequences, and cut the IJRR and MVSEC sequences with boundaries as listed in Table 4. The same MVSEC sequence cuts are used for optical flow evaluation.

Table 4: Cut boundaries for IJRR and MVSEC sequences.

| IJRR | | | MVSEC | | |
|---|---|---|---|---|---|
| Sequence | Start [s] | End [s] | Sequence | Start [s] | End [s] |
| boxes_6dof | 5.0 | 20.0 | indoor_flying1 | 10.0 | 70.0 |
| calibration | 5.0 | 20.0 | indoor_flying2 | 10.0 | 70.0 |
| dynamic_6dof | 5.0 | 20.0 | indoor_flying3 | 10.0 | 70.0 |
| office_zigzag | 5.0 | 12.0 | indoor_flying4 | 10.0 | 19.8 |
| poster_6dof | 5.0 | 20.0 | outdoor_day1 | 0.0 | 60.0 |
| shapes_6dof | 5.0 | 20.0 | outdoor_day2 | 100.0 | 160.0 |
| slider_depth | 1.0 | 2.5 | | | |

The frames provided in the EVAID [6] dataset have relatively better visual quality. To reduce the testing burden, we only crop a 5-second segment (up to 750 frames, as EVAID has a high frame rate) from each of the longer sequences. The boundaries of these segments are listed in Table 5.

Table 5: Cut boundaries for EVAID sequences.

| EVAID | | | | | | | |
|---|---|---|---|---|---|---|---|
| Sequence | Start [s] | End [s] | Frames | Sequence | Start [s] | End [s] | Frames |
| ball | 0.0 | 5.0 | 500 | bear | 0.0 | 2.7 | 66 |
| box | 0.0 | 5.0 | 100 | building | 0.0 | 5.0 | 750 |
| outdoor | 0.0 | 1.4 | 26 | playball | 25.0 | 30.0 | 750 |
| room1 | 0.0 | 5.0 | 750 | sculpture | 0.0 | 5.0 | 750 |
| toy | 0.0 | 5.0 | 100 | traffic | 0.0 | 5.0 | 500 |
| wall | 0.0 | 5.0 | 750 | | | | |

We did not include the "blocks" and "umbrella" sequences, because the cameras were completely static and no signal events were triggered in the background throughout the sequences. As a result, event-based video reconstruction methods can only wildly guess the backgrounds, which causes metrics to be highly random. We cut from the later part of the "playball" sequence for the same

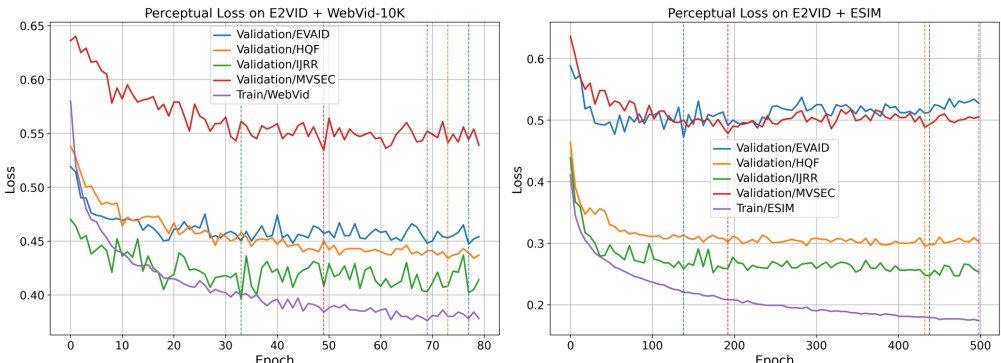

Figure 7: Performance on real data fluctuates as training loss decreases.

reason: there was no camera motion at the beginning. We exclude the "room2" sequence because it is highly homogeneous with the "room1" sequence.

We observe that, although the loss on the synthetic training datasets gradually converges after sufficient epochs, the test performances on real datasets still exhibit considerable instability. Figure 7 shows the train loss and validation loss curves of E2VID in the experiments on WebVid10K ((d) in Table 9) and ESIM ((l) in Table 9). The checkpoints achieving the best performance on different datasets are often not the same, and the choice of checkpoint for final testing has a significant impact on performance evaluation. In our study, for all experiments, we perform approximately the same number of validation epochs on real datasets throughout training. For example, for E2VID-ESIM trained over 500 epochs, validation is performed every 6 epochs; for E2VID-WebVid10K trained over 80 epochs, every 1 epoch; for E2VID-WebVid1K trained over 800 epochs, every 10 epochs; and for E2VID-WebVid100 trained over 8000 epochs, every 100 epochs. Then, we select the checkpoint corresponding to the lowest average LPIPS loss across the HQF and EVAID datasets as the final model for testing.

**License.**    MVSEC [43] is released under the CC BY-SA 4.0 license. IJRR [25] is released under the CC BY-NC-SA 3.0 license. HQF [33] and EVAID [6] did not include a license.

# B    Training Details

All our experiments were conducted on single-GPU instances, and the peak GPU memory usage does not exceed 80 GB. We followed the optimizer and learning rate settings of the original models.

**E2VID.**    The E2VID models were trained with a batch size of 12. The optimizer is Adam with a constant learning rate of 0.0001, no weight decay, and AMSGrad enabled.

**ETNet.**    The ETNet models were trained with a batch size of 6. The optimizer is AdamW with initial learning rate 0.0002, weight decay rate 0.01, and AMSGrad enabled. We adjust the gamma value of the exponential learning rate scheduler to 0.94 since we only train for 100 epochs.

**HyperE2VID.**    The HyperE2VID models were trained with a batch size of 12. The optimizer is Adam with a constant learning rate of 0.001, no weight decay, and AMSGrad enabled.

We trained the models on datasets rendered by ESIM [28] as well as the WebVid [2] dataset. As shown in Figure 8, scenes from the WebVid dataset are far more diverse and realistic than the flying-image style scenes rendered by ESIM, which is the source of our models' exceeding performance.

**License.**    WebVid [2] states on its website (https://github.com/m-bain/webvid) that the dataset can be used for non-commercial purposes.

# C    Dataset Quality Discussion

The low quality of the APS frames in IJRR and MVSEC has already been reported by Stoffregen *et al.* [33], who selected relatively good clips from IJRR and MVSEC to evaluate on. However, we observe that even the selected clips are far from ideal.

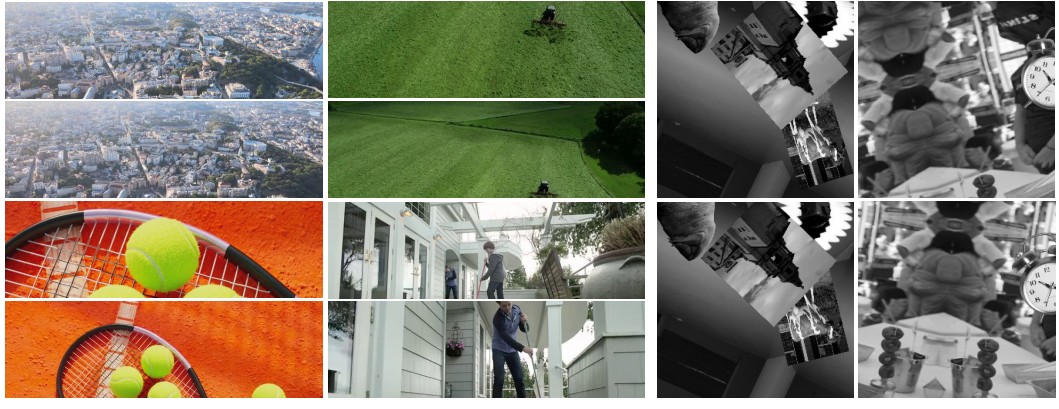

(a) WebVid scenes                    (b) ESIM scenes

Figure 8: Scenes from the WebVid dataset are much more diverse and realistic than the flying-image style scenes rendered by ESIM [33].

For example, as shown in Figure 9, the ground truth frame from `IJRR/dynamic_6dof` is underexposed and too dark. Even after adjusting the image, the person's leg under the table cannot be seen. The image also suffers from vignetting. Hence, the GT-based metrics of E2VID methods will be punished if they generate frames with moderate exposure or successfully reconstruct the details under the table, causing the metrics to be misleading.

The ground truth frame example from `MVSEC/outdoor_day1` is also underexposed. Moreover, when we adjust the frame to be brighter, we can observe that it is very noisy and unsuitable to serve as ground truth.

The quality of the event pixels also varies across datasets. In Figure 10, for sequences from MVSEC, IJRR, HQF, and EVAID, we visualize the total number of events triggered on each pixel within the sequence.

In all DAVIS datasets (MVSEC, IJRR, HQF), we can observe "hot pixels" (dark red dots) that keep omitting events. We also observe vertical stripe-like patterns, suggesting variations in threshold levels between odd and even pixel columns.

In the `MVSEC/indoor_flying` sequences, we also observe many "dead pixels" (dark blue particles) that do not trigger any events despite the large camera motion. This may have caused the dirty-window artifacts (fixed patterns that hover on top of videos, like looking through a dirty window) shared by the reconstructions of almost all the methods.

We did not observe these problems in the EVAID dataset, which was captured with a Prophesee EKV4 event camera. Hence, we speculate that these issues are inherent to the DAVIS camera series. While we could simulate corresponding defects during event synthesis to improve model generalization on DAVIS data, the fact that these cameras are no longer available on the market makes backward compatibility less valuable. Therefore, we have decided to look ahead and directly update our testing benchmark to focus on more advanced camera models.

## D   Ablation Study Results

### D.1   Dataset quality

In the WebVid dataset, a large proportion of videos have out-of-focus blur due to large aperture settings, and some videos are synthetic CG instead of real videos. We tried to improve dataset quality by filtering them out, but the results were counterintuitive: using the unfiltered version produced better results.

Specifically, from the 2.5M videos of WebVid, we arbitrarily selected a subset of 32K videos and labeled them with the large vision language model Qwen2.5-VL [1]. For each video, we sample one frame and ask Qwen2.5-VL if there is any post-production artefact (such as subtitles/CG/...) in the image, if the image is blank, and if the image has out-of-focus or motion blur. We exclude

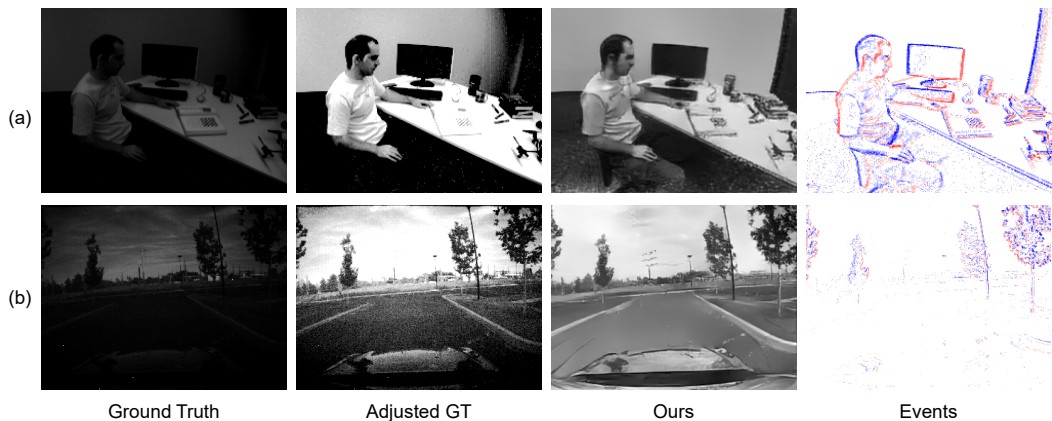

Ground Truth      Adjusted GT      Ours      Events

Figure 9: Frames from IJRR (a) and MVSEC (b) are far from ideal.

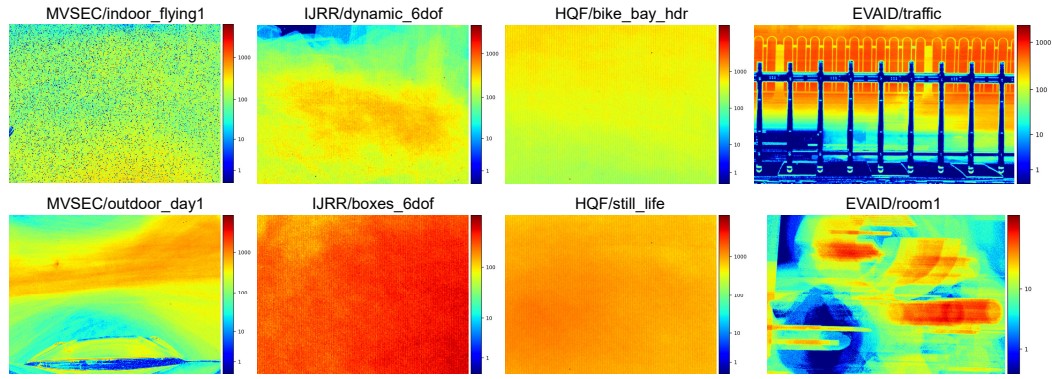

Figure 10: Event count maps from real event data sequences.

all videos with post-production artefacts, blank content, or blurriness. From the remainings, we randomly sampled 10K videos, forming the dataset WebVid10K*.

By comparing lines (d) and (n) in Table 9, we see that models trained on WebVid10K perform slightly better than on WebVid10K*. This may be because the filtering decreases dataset diversity, since specific sample types, such as daytime outdoor long-shot videos, are less likely to be filtered out. The semantic statistics of the two datasets (annotated using Qwen2.5-VL [1]) are listed in Table 6.

Table 6: Semantic statistics of WebVid datasets.

| Description | WebVid10K* (%) | WebVid10K (%) |
|---|---|---|
| Is real video | 100.00 | 76.83 |
| From outdoor scene | 70.63 | 53.15 |
| From indoor scene | 21.33 | 25.09 |
| Is daytime | 75.53 | 60.51 |
| Is nighttime | 3.44 | 9.56 |
| Has water | 22.22 | 16.17 |
| Has humans | 30.09 | 34.11 |
| Has sky | 46.19 | 27.59 |
| Is blank | 0.00 | 3.44 |
| Has defocus blur | 0.00 | 41.10 |
| Has motion blur | 0.00 | 9.32 |
| Contains text | 3.76 | 3.43 |

To demonstrate the importance of exposure quality, we intentionally degenerate the dynamic range of the video dataset, and observe its effect on the model. Specifically, in each iteration, with 80%

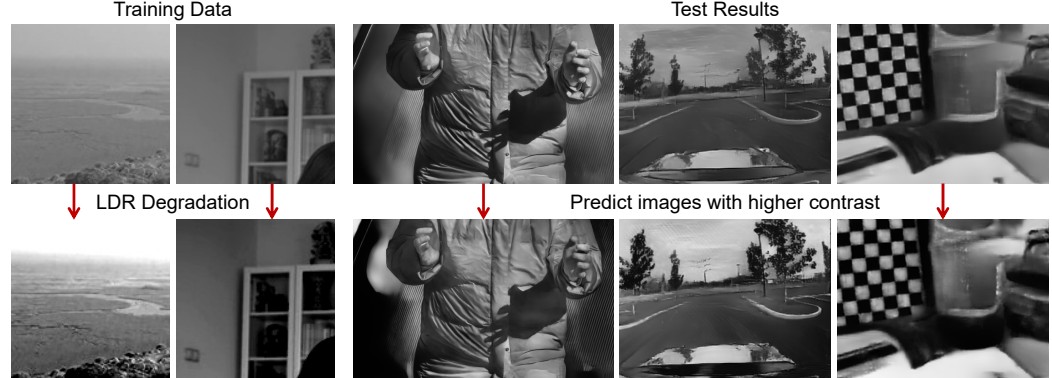

Figure 11: Degenerating the dynamic range of training videos makes the model reconstruct images with higher contrast.

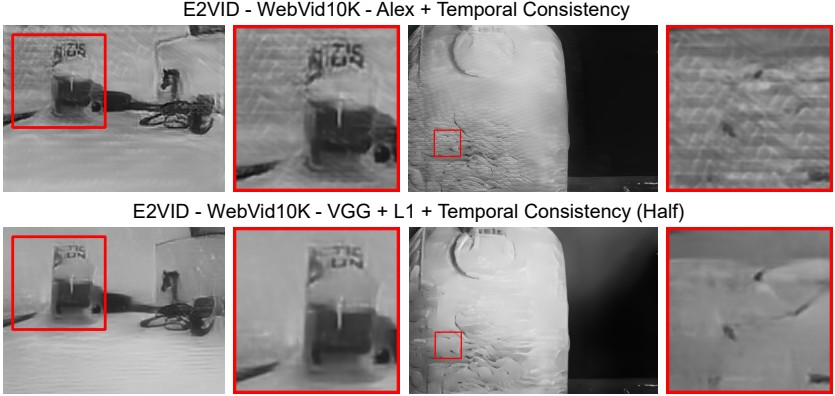

Figure 12: The Alex LPIPS loss introduces arrow-like patterned artefacts when the video degenerates.

probability, we degenerate the video frames with a random scale $s$ between 1 and 3:

$$F_{\text{degrade}} = \text{Clip}((F - 127.5) * s + 127.5,\ 0,\ 255). \tag{9}$$

This increases the contrast of the video, and creates more overexposed and underexposed areas. Our experiments show that the corresponding model indeed learns to reconstruct images in a more high contrast style, as shown in Figure 11. The quantitative metrics are listed in line (o) of Table 9.

## D.2 Design of loss function

**Alex or VGG for perceptual loss.** We observe that using the Alex model for the computation of LPIPS loss introduces a certain type of patterned artefact when events are sparse and the outputs start to degenerate. These patterns are visually unpleasant, and they disappear when we change to using the VGG LPIPS loss (combined with L1 loss). In the ablation experiment (p) of Table 9, we train on WebVid10K using the original loss combination, Alex LPIPS + full temporal consistency (with optical flow from RAFT-Small [32]). Figure 12 shows the unpleasant arrow-like patterns, and the corresponding metrics in the table are also worse than experiment (d), which was trained on our loss design.

**Pseudo ground truth optical flow.** Model-based event simulators such as ESIM provide ground truth optical flow, which is used to compute temporal consistency loss. When we shift to video-based simulation, ground truth optical flow is no longer accessible. Although we can predict optical flow using the image frames, we worry that the errors in prediction may have negative effects on training.

In experiment (t) of Table 9, we train the E2VID model using predictions of RAFT-Small [32] as optical flow. The experiment (l) of Table 9 has the same setting, except for the optical flow used in the temporal consistency loss is ESIM-rendered ground truth. Through comparison, no significant negative influence can be observed.

**Half temporal consistency loss.**    In this paper, we introduce half temporal consistency loss, *i.e.*, only computing temporal consistency loss over the last 20 frames of a 40-frame sequence. In Table 9, experiments (r) (s) (t) explore the benefits of using half temporal consistency loss instead of no temporal consistency loss (only Alex LPIPS, (r)) or full temporal consistency loss (Alex+TC-RAFT, (t)). Although the half consistency loss experiment (s) do not exceed in the frame-based quantitative metrics, the elimination of flickering and the relief of dirty-window artefacts can be observed in the reconstructed videos, which can be found in the supplementary video.

## D.3   Comparison against the V2E solution.

In order to compare our V2V framework to a "frame interpolation + event simulation" approach, we interpolated the WebVid-100 dataset by 8x with PerVFI [37] and used the popular V2E simulator [13] to generate training events. We used the "clean" noise mode. For the thresholds, we linearly scaled the random thresholds in the "fixed thresholds" ablation experiment due to the different logarithm mapping of V2E from ESIM (our V2V framework follows ESIM).

In the simulation, a video (1066695484.mp4, with original size 1.2MB) produced 14GB of events. This caused our post-processing step to fail due to out-of-memory issues. We discarded this sample, leaving us with a WebVid-99 dataset. The WebVid-99 dataset (in raw events) sums to 12 GB space, while the original WebVid-100 videos only sum to 210 MB.

We then stacked the events into interpolated voxel bins. In this stage, we need to select the durations of the voxel bins. In the V2V experiments, each voxel corresponds to the events between 6 real frames, and the E2VID models reconstruct 24/5 FPS videos. We tried two binning strategies in our V2E experiments:

In strategy 1, we encode all the events between 1+1 original frames (crossing 7 interpolated frames) into a single voxel. This makes the E2VID models reconstruct 24 FPS videos, utilizing the high temporal resolution from frame interpolation. We annotate this as "V2E-1".

In strategy 2, we encode all the events between 5+1 original frames (crossing 35 interpolated frames) into a single voxel. This makes the E2VID models reconstruct 24/5 FPS videos, aligning to the V2V experiments. We annotate this as "V2E-5".

During training, we used the same augmentation noise (gaussian noise + hot pixels) as the ESIM protocol. For the V2E-5 experiment, we train for 8000 epochs, aligning to the V2V-WebVid-100 experiment. For the V2E-1 experiment, it produces 5x more training samples, so we train for 1600 epochs.

The test results are shown in Table 7. The V2V-1 experiment produced better results than the V2V-5 experiment, showing that training on shorter bins may indeed benefit model performance. However, both V2E experiments produced metrics worse than the "WebVid-100-Fixed" experiment, which also uses fixed thresholds for each video. This shows that our V2V method not only exceeds the V2E approach in efficiency, but also produces models with better performance.

Table 7: Comparison of V2V solution and V2E approach. Lower (↓) MSE and LPIPS values and higher (↑) SSIM values are desirable. We highlight the best values with green .

| Model | Train Dataset | Loss | Eps | HQF | | | EVAID | | |
|-------|---------------|------|-----|------|-------|--------|------|-------|--------|
| | | | | MSE↓ | SSIM↑ | LPIPS↓ | MSE↓ | SSIM↑ | LPIPS↓ |
| E2VID | V2V-100-Fixed | V+L1+TH | 8000 | 0.049 | 0.611 | 0.330 | 0.073 | 0.604 | 0.451 |
| E2VID | V2V-100 | V+L1+TH | 8000 | 0.042 | 0.626 | 0.327 | 0.078 | 0.594 | 0.441 |
| E2VID | WebVid99-V2E-1 | V+L1+TH | 1600 | 0.057 | 0.524 | 0.459 | 0.116 | 0.484 | 0.561 |
| E2VID | WebVid99-V2E-5 | V+L1+TH | 8000 | 0.077 | 0.470 | 0.569 | 0.117 | 0.483 | 0.588 |

## D.4   Finetuning on test datasets.

Our V2V framework produces models with strong zero-shot generalization abilities. In order to show this, we conducted a finetuning experiment on EVAID to show that our zero-shot model can outperform a model trained or finetuned with limited real event data.

Our EVAID test set has 10 sequences. We took 7 sequences (Ball, Box, Building, Outdoor, Playball, Room1, Toy, Wall) as a train set, and 3 sequences (Bear, Sculpture, Traffic) as the new test set.

**Zero-shot inference.** We directly used the V2V-E2VID model weights, trained on the WebVid dataset, for zero-shot inference on the EVAID test set.

**Finetuning on EVAID.** We finetuned the pretrained V2V-E2VID model on the EVAID train set for 100 epochs, with a learning rate of 0.0001, observing that the validation loss had converged.

**Directly training on EVAID.** We trained an E2VID model from scratch on the EVAID train set. We trained it for 2000 epochs with learning rate 0.0001, observing that the validation loss had converged.

The resulting metrics of the three models are shown in Table 8. Surprisingly, the zero-shot model performed best. Finetuning caused a slight performance drop, while training on real data from scratch performed worst. This is likely due to the small size of the EVAID dataset, which caused the model to overfit to the training sequences and fail to generalize to the test sequences.

Table 8: E2VID directly trained or finetuned with EVAID. Lower (↓) MSE and LPIPS values and higher (↑) SSIM values are desirable. We highlight the best values with green .

| Model | Train Dataset | Loss | Eps | EVAID | | |
|---|---|---|---|---|---|---|
| | | | | MSE↓ | SSIM↑ | LPIPS↓ |
| E2VID | V2V-10K (Zero-shot) | V+L1+TH | 80 | 0.056 | 0.661 | 0.424 |
| E2VID | V2V-10K + EVAID (Finetuned) | V+L1+TH | 80 + 100 | 0.064 | 0.645 | 0.458 |
| E2VID | EVAID (From scratch) | V+L1+TH | 2000 | 0.078 | 0.483 | 0.658 |

## D.5 Performance gain on downstream tasks

In order to demonstrate the reusability of the V2V weights on downstream tasks, we conducted an experiment on the N-MNIST classification task.

The MNIST dataset is an image-based dataset, corresponding to the task of classifying hand-written digits, frequently used as a toy dataset in computer vision. N-MNIST [26] is a corresponding event-based classification dataset, recorded by saccading the original MNIST images with an real event camera. We chose this task due to its convenience: the dataset is small and code is simple.

We first trained a very simple convolution model on the train split of MNIST (the image version). On the test split (images), the model achieved an accuarcy of 98.76%.

Then we explored using video reconstruction models in a zero-shot pipeline for the N-MNIST task. The pipeline is as follows:

- Take real event streams from the N-MNIST test split. Stack them to 5*5 event bins.
- For each of the 5 voxels (each with 5 bins), use the E2VID model to reconstruct an image frame. This produces 5 reconstructed frames.
- Use the last frame as the input to the image-based MNIST classifier, and predict the digit.

When using the original E2VID model, the resulting accuracy on the N-MNIST test set is 47.46%. When changing to the V2V-E2VID model, the accuracy improves to 63.62%. This is due to the better reconstruction quality of the V2V-E2VID model.

This shows that the quality improvement brought by the V2V framework can be utilized by zero-shot downstream tasks. Note that there are existing classification models that can generalize much better; the purpose of using a weak model is to compare the original E2VID against V2V-E2VID.

## E   Additional Reconstruction Results

We provide qualitative results of our method V2V-E2VID (trained on WebVid10K) in Figure 13, Figure 14, Figure 15 and Figure 16. Compared to the baseline method E2VID+, our method produces fewer artefacts and better image contrasts.

In Table 9, we report quantitative metrics over the datasets EVAID, HQF, IJRR, and MVSEC. Quantitative metrics of all ablation studies are also provided.

Table 9: Qualitative results of all experiments on all datasets. Duplicated lines are provided for convenience of comparison.

| Idx | Model | Training Dataset | Loss | Epochs | HQF | | | EVAID | | | IJRR | | | MVSEC | | |
|---|---|---|---|---|---|---|---|---|---|---|---|---|---|---|---|---|
| | | | | | MSE | SSIM | LPIPS | MSE | SSIM | LPIPS | MSE | SSIM | LPIPS | MSE | SSIM | LPIPS |
| | | | | | **V2V - E2VID** | | | | | | | | | | | |
| (a) | E2VID | Pretrained (ESIM) | Alex+TC | 500 | 0.037 | 0.638 | 0.256 | 0.088 | 0.526 | 0.469 | 0.063 | 0.556 | 0.238 | 0.132 | 0.34 | 0.506 |
| (b) | E2VID | WebVid100 | V+L1+TH | 8000 | 0.042 | 0.626 | 0.327 | 0.078 | 0.594 | 0.441 | 0.067 | 0.55 | 0.278 | 0.076 | 0.385 | 0.525 |
| (c) | E2VID | WebVid1K | V+L1+TH | 800 | 0.037 | 0.661 | 0.286 | 0.072 | 0.627 | 0.405 | 0.078 | 0.569 | 0.251 | 0.103 | 0.36 | 0.509 |
| (d) | E2VID | WebVid10K | V+L1+TH | 80 | 0.032 | 0.676 | 0.265 | 0.054 | 0.663 | 0.371 | 0.07 | 0.562 | 0.248 | 0.066 | 0.406 | 0.465 |
| (i) | E2VID | WebVid100-F | V+L1+TH | 8000 | 0.049 | 0.611 | 0.33 | 0.073 | 0.604 | 0.451 | 0.077 | 0.554 | 0.287 | 0.096 | 0.356 | 0.538 |
| (j) | E2VID | WebVid1K-F | V+L1+TH | 800 | 0.039 | 0.64 | 0.293 | 0.066 | 0.617 | 0.413 | 0.1 | 0.529 | 0.275 | 0.104 | 0.344 | 0.513 |
| (k) | E2VID | WebVid10K-F | V+L1+TH | 80 | 0.032 | 0.666 | 0.264 | 0.058 | 0.644 | 0.378 | 0.086 | 0.539 | 0.256 | 0.097 | 0.37 | 0.492 |
| | | | | | **V2V - Other models** | | | | | | | | | | | |
| (e) | HyperE2VID | Pretrained (ESIM) | Alex+TC | 400 | 0.032 | 0.646 | 0.265 | 0.071 | 0.533 | 0.474 | 0.036 | 0.624 | 0.227 | 0.07 | 0.419 | 0.473 |
| (f) | HyperE2VID | WebVid10K | V+L1+TH | 60 | 0.035 | 0.671 | 0.269 | 0.055 | 0.654 | 0.377 | 0.059 | 0.6 | 0.231 | 0.077 | 0.34 | 0.57 |
| (g) | ETNet | Pretrained (ESIM) | Alex+TC | 700 | 0.035 | 0.642 | 0.274 | 0.08 | 0.541 | 0.523 | 0.049 | 0.59 | 0.236 | 0.111 | 0.36 | 0.491 |
| (h) | ETNet | WebVid10K | V+L1+TH | 100 | 0.039 | 0.641 | 0.306 | 0.056 | 0.61 | 0.406 | 0.085 | 0.544 | 0.26 | 0.103 | 0.351 | 0.551 |
| | | | | | **Ablation - Voxel Representation** | | | | | | | | | | | |
| (l) | E2VID | ESIM-280-Intp. | Alex+TC | 500 | 0.041 | 0.599 | 0.306 | 0.078 | 0.526 | 0.478 | 0.061 | 0.561 | 0.258 | 0.127 | 0.324 | 0.49 |
| (m) | E2VID | ESIM-280-Disc. | Alex+TC | 500 | 0.04 | 0.608 | 0.296 | 0.062 | 0.548 | 0.45 | 0.06 | 0.57 | 0.255 | 0.125 | 0.345 | 0.494 |
| | | | | | **Ablation - Dataset quality** | | | | | | | | | | | |
| (d) | E2VID | WebVid10K | V+L1+TH | 80 | 0.032 | 0.676 | 0.265 | 0.054 | 0.663 | 0.371 | 0.07 | 0.562 | 0.248 | 0.066 | 0.406 | 0.465 |
| (n) | E2VID | WebVid10K* | V+L1+TH | 80 | 0.038 | 0.662 | 0.275 | 0.059 | 0.645 | 0.389 | 0.069 | 0.574 | 0.238 | 0.072 | 0.398 | 0.468 |
| (o) | E2VID | WebVid10K-LDR | V+L1+TH | 80 | 0.034 | 0.647 | 0.287 | 0.09 | 0.565 | 0.44 | 0.092 | 0.56 | 0.265 | 0.092 | 0.386 | 0.48 |
| | | | | | **Ablation - Loss design** | | | | | | | | | | | |
| (d) | E2VID | WebVid10K | V+L1+TH | 80 | 0.032 | 0.676 | 0.265 | 0.054 | 0.663 | 0.371 | 0.07 | 0.562 | 0.248 | 0.066 | 0.406 | 0.465 |
| (p) | E2VID | WebVid10K | Alex+TC | 80 | 0.034 | 0.652 | 0.249 | 0.054 | 0.604 | 0.388 | 0.082 | 0.519 | 0.268 | 0.088 | 0.375 | 0.453 |
| (r) | E2VID | ESIM-280-Intp. | Alex | 500 | 0.041 | 0.621 | 0.285 | 0.06 | 0.55 | 0.457 | 0.057 | 0.584 | 0.249 | 0.119 | 0.368 | 0.497 |
| (s) | E2VID | ESIM-280-Intp. | Alex+TH-RAFT | 500 | 0.045 | 0.608 | 0.285 | 0.072 | 0.549 | 0.448 | 0.083 | 0.533 | 0.266 | 0.165 | 0.313 | 0.532 |
| (t) | E2VID | ESIM-280-Intp. | Alex+TC-RAFT | 500 | 0.039 | 0.613 | 0.276 | 0.07 | 0.541 | 0.451 | 0.071 | 0.552 | 0.255 | 0.141 | 0.335 | 0.504 |
| (l) | E2VID | ESIM-280-Intp. | Alex+TC-GT | 500 | 0.041 | 0.599 | 0.306 | 0.078 | 0.526 | 0.478 | 0.061 | 0.561 | 0.258 | 0.127 | 0.324 | 0.49 |

## F   Qualitative Optical Flow Results

In Figure 17, we provide visualizations of our optical flow predictions. Our retrained model V2V-EvFlowNet produces denser results and fewer black hollows.

## G   Broader Impacts

**Positive impacts.**   Our V2V module enables scaling up event-based datasets with less storage usage, less data transfer burden, more sample diversity, and greater augmentation flexibility. This will lower the economic barriers for conducting event-based vision research, empower more researchers to advance related work, and leverage the event camera's high temporal resolution, high dynamic range, and low power consumption to push the boundaries of machine vision.

**Negative Impacts.**   Video-based training enables models to learn prior knowledge from the video data. As a result, they may suffer from biased or discriminatory features present in the video datasets, which calls for future researchers to more carefully consider the data used for training.

E2VID+ (Baseline)          V2V-E2VID-10K (Ours)          Ground Truth

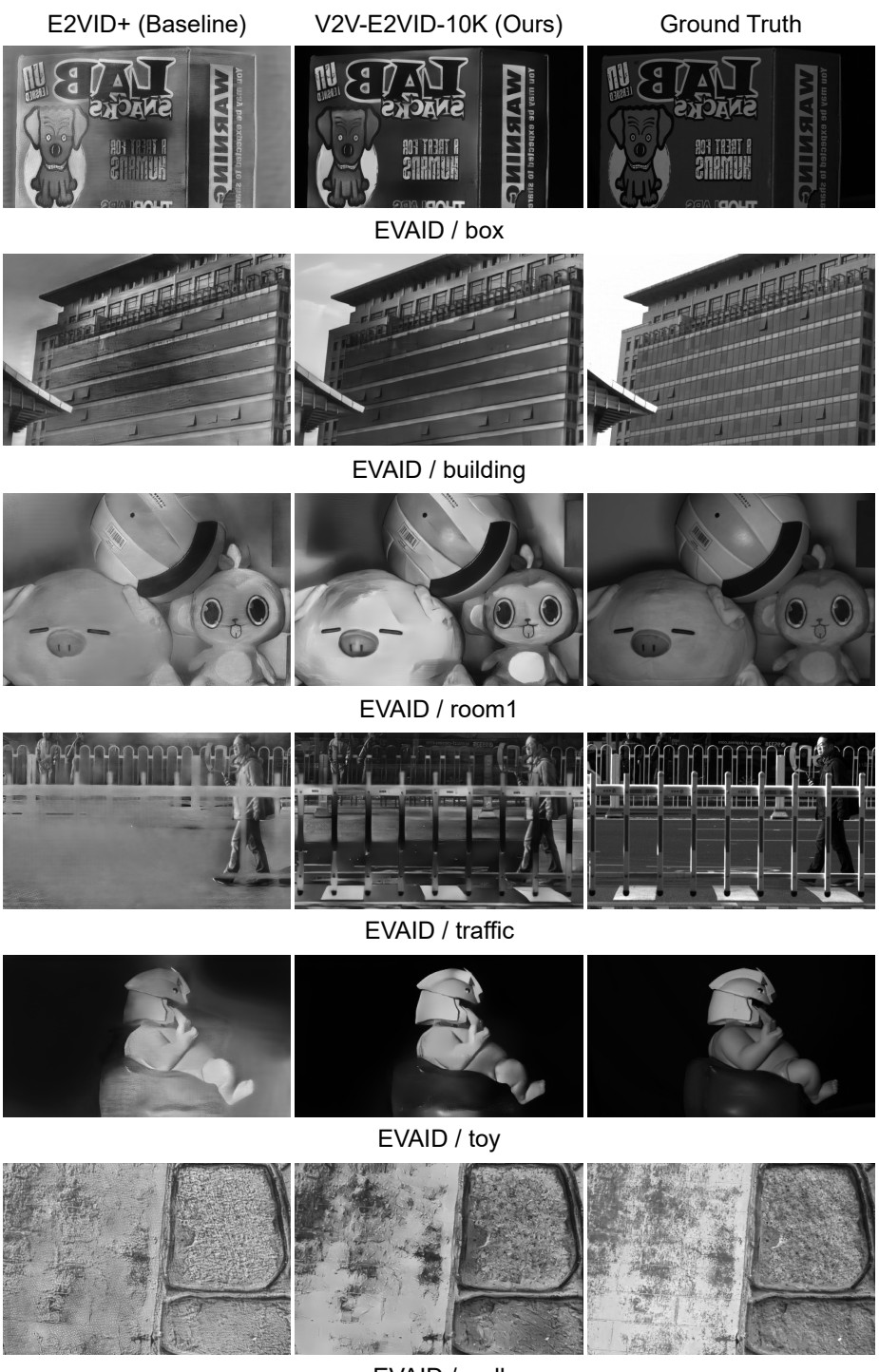

EVAID / box

EVAID / building

EVAID / room1

EVAID / traffic

EVAID / toy

EVAID / wall

Figure 13: Qualitative results on the EVAID dataset.

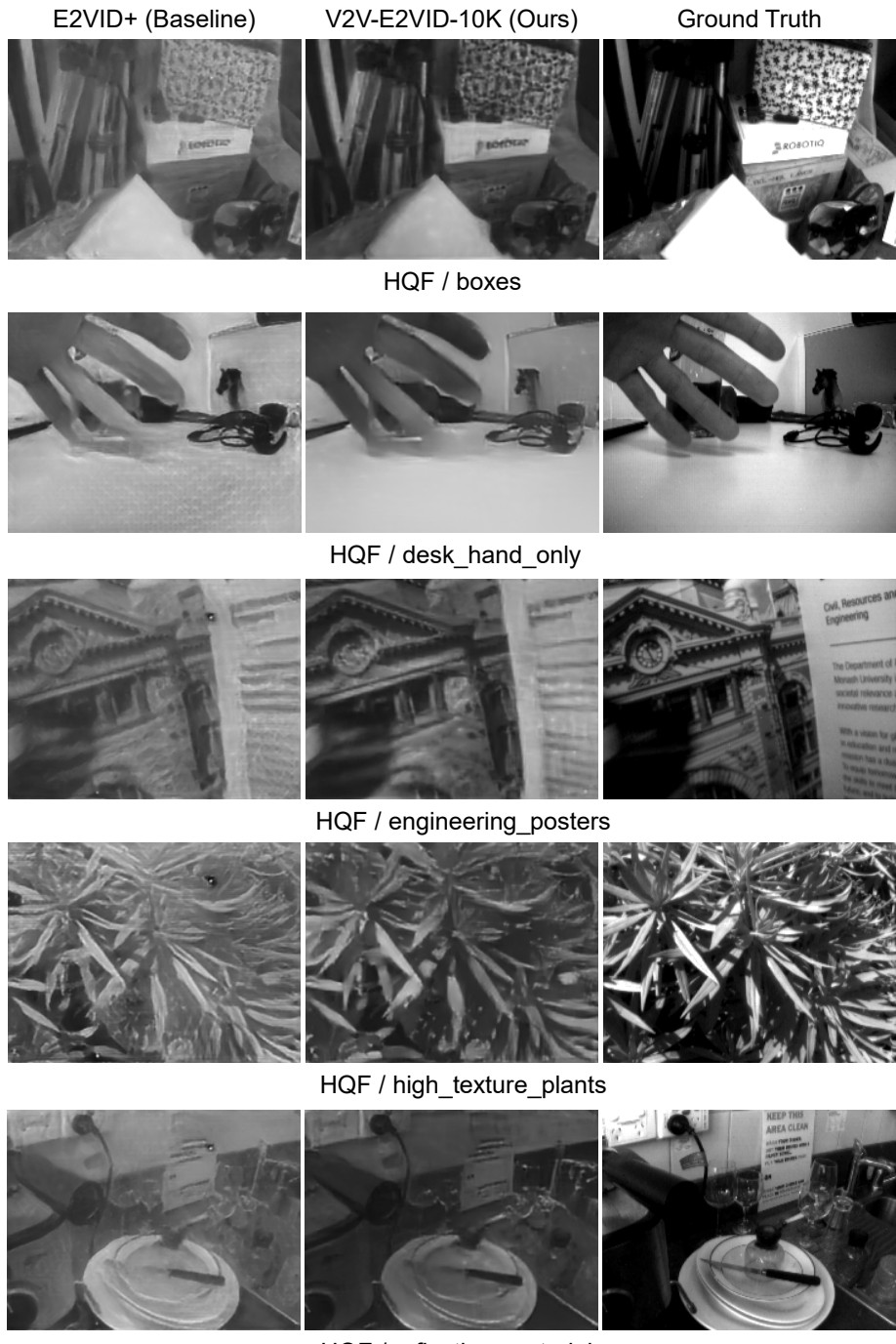

Figure 14: Qualitative results on the HQF dataset.

E2VID+ (Baseline)      V2V-E2VID-10K (Ours)      Ground Truth

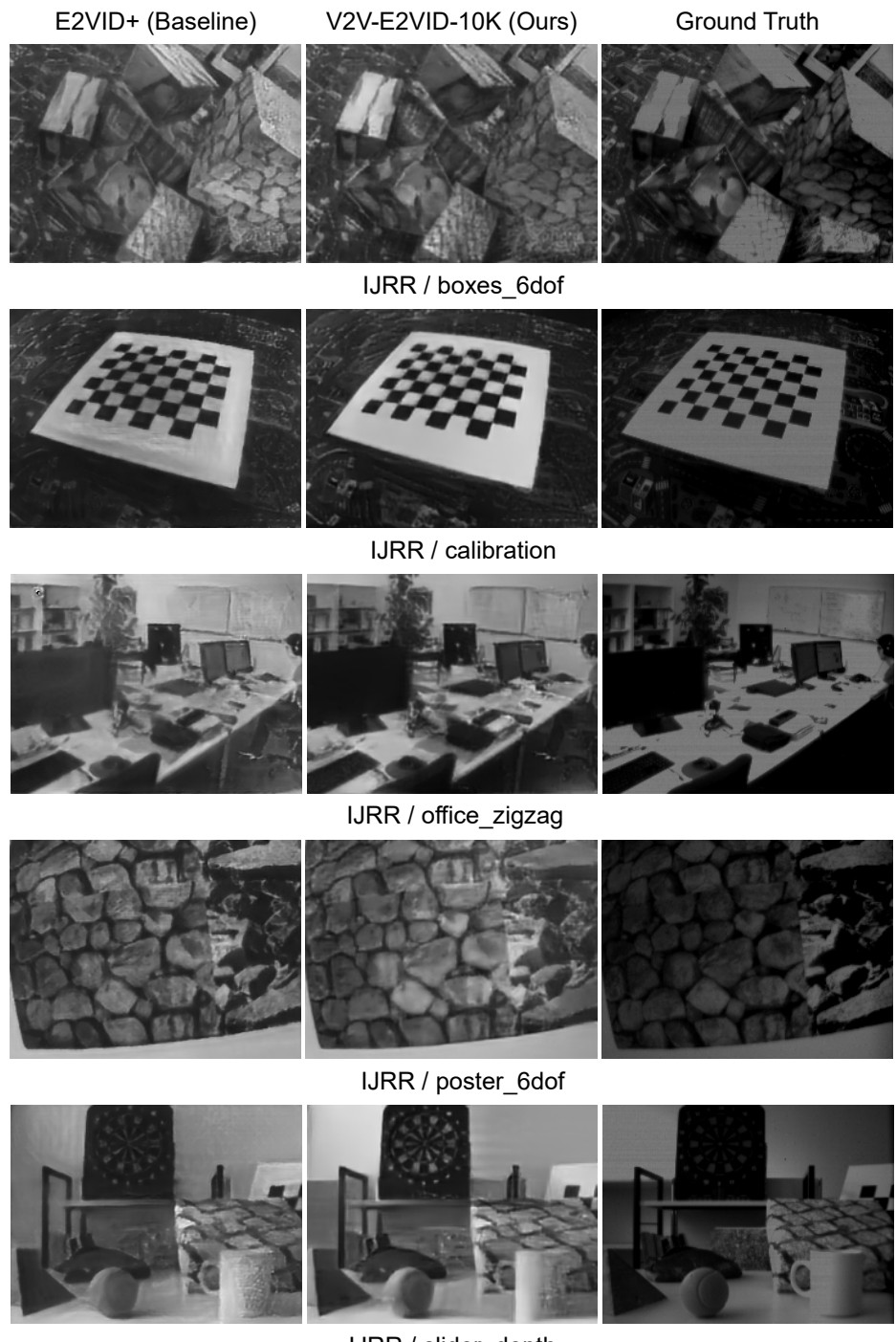

IJRR / boxes_6dof

IJRR / calibration

IJRR / office_zigzag

IJRR / poster_6dof

IJRR / slider_depth

Figure 15: Qualitative results on the IJRR dataset.

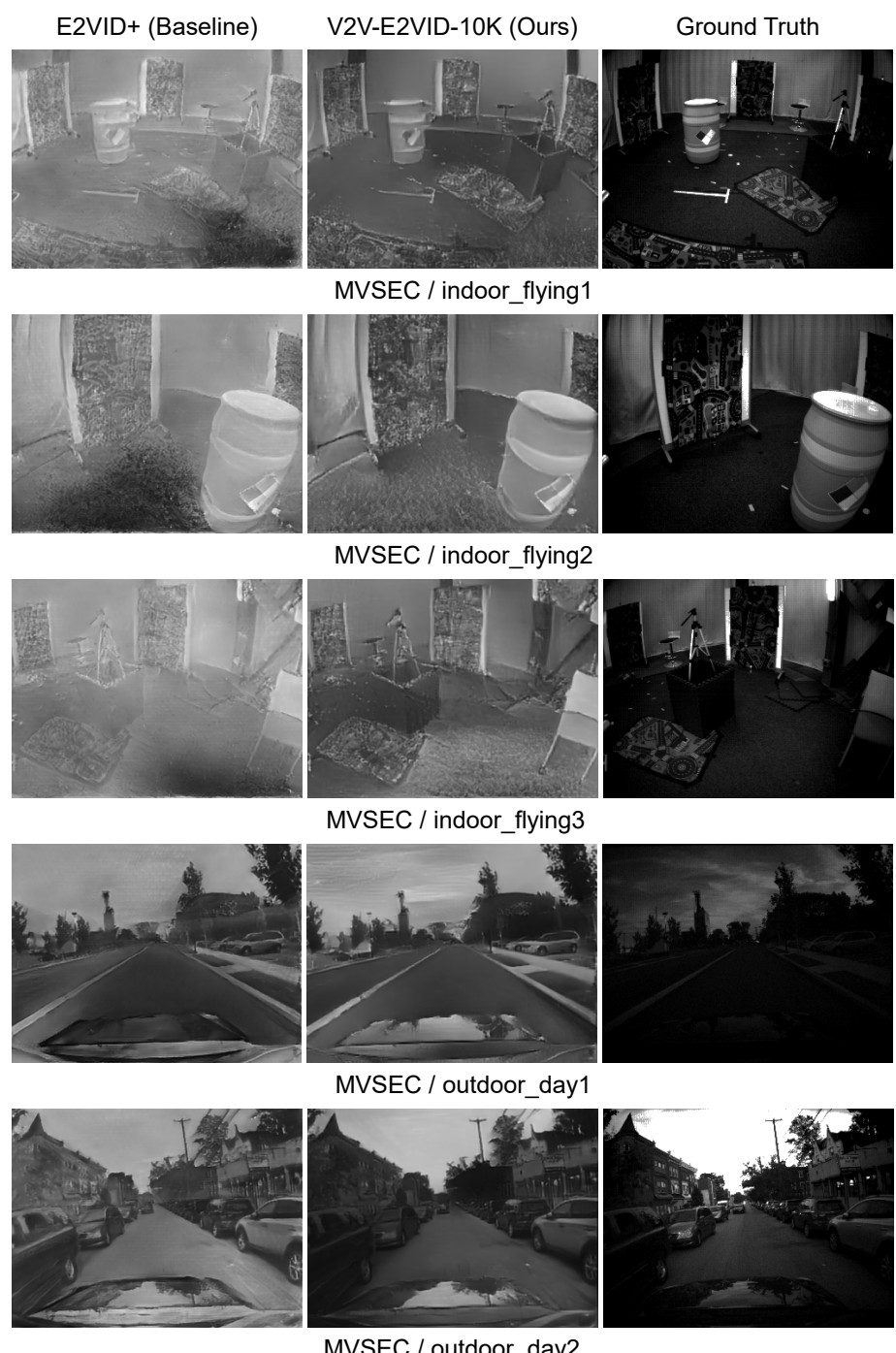

Figure 16: Qualitative results on the MVSEC dataset.

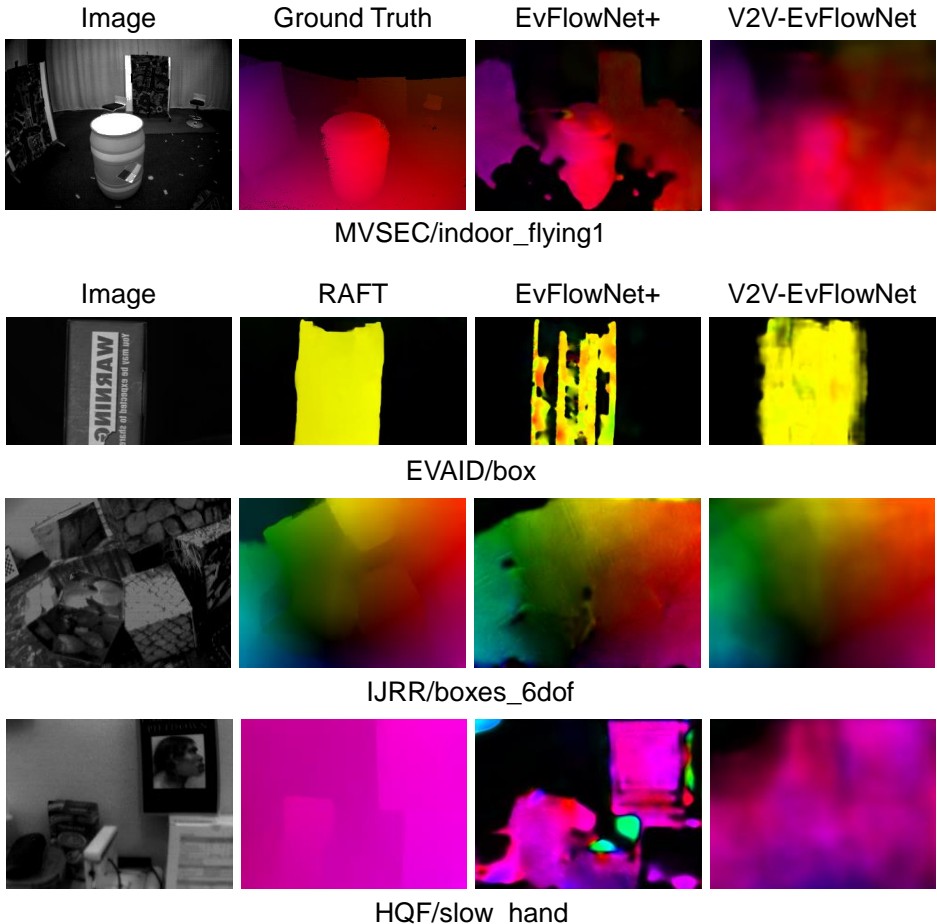

Figure 17: Qualitative results of V2V-EvFlow.

