# OpenReview forum: "V2V: Scaling Event-Based Vision through Efficient Video-to-Voxel Simulation"
_NeurIPS.cc/2025/Conference — NeurIPS 2025 poster_

### Official Review · Reviewer_bJ8Q · 2025-06-25

**Clarity:** 3
**Significance:** 3
**Originality:** 3
**Rating:** 4
**Confidence:** 5

**Summary:**

This paper investigates the potential of using video-based voxel generation to significantly enhance the training efficiency of event-to-video (E2V) models. The authors also introduce a large-scale synthetic dataset to facilitate this process.

**Questions:**

1.	My primary concern is that most of the experiments evaluate the joint effect of the V2V method and large-scale synthetic data. How can the effectiveness of the V2V approach itself be isolated and validated independently?

2.	The performance gains on the EVAID dataset are significantly greater than those on the HQF dataset. In particular, the improvements in HQF are marginal. The authors should provide a more detailed discussion of the underlying reasons for this discrepancy.

**Ethical Concerns:**

["NO or VERY MINOR ethics concerns only"]

**Final Justification:**

The author's rebuttal has largely addressed my concerns.

**Limitations:**

The authors did not discuss the limitations of the techniques in the paper.

**Quality:**

3

**Strengths And Weaknesses:**

Strengths:

1. The construction of a large-scale, multi-scene event dataset is valuable, especially given the current limitations in real-world data.

2. The paper addresses critical issues in event frame reconstruction, including storage and computational efficiency, from a novel and interesting perspective.


Weaknesses:
1. Converting video to event voxel representations is similar to the first stage of V2CE. While V2CE does not thoroughly explore this video-to-voxel conversion, a clearer comparison and discussion are necessary to clarify the novelty and contribution of the proposed method.

    [1] Zhang Z, Cui S, Chai K, et al. V2ce: Video to continuous events simulator[C]//2024 IEEE International Conference on Robotics and Automation (ICRA). IEEE, 2024: 12455-12461.

2. The paper does not further explore the model design for voxel-to-frame translation, which is unfortunate and significantly limits the technical depth of the work.

3. The proposed dataset is synthetic. While large-scale, the lack of a large, real-world event dataset remains a key bottleneck in the field, and this work does not address that gap.

4. While the authors demonstrate consistent improvements through pretraining on the proposed WebVid synthetic dataset, there is insufficient ablation or analysis to disentangle the benefits of the V2V approach from the benefits of large-scale pretraining.

---

> ### Author Rebuttal · Authors · 2025-07-31
>
> **Response to Reviewer bJ8Q**
>
> Thank you for your detailed feedback. We will address your concerns below.
>
> **S1. Clarification of contributions.**
>
> Thank you for your appreciation of our contributions. However, we would like to clarify that V2V is not a specific dataset, but a general framework for efficient event data simulation, especially for large scale datasets. It can be used to generate synthetic data for a large range of event-based vision tasks, including but not limited to video reconstruction and **optical flow estimation**.
>
> **W1. Similarities to existing simulators.**
>
> It is indeed common for event simulators to first calculate the amount of events triggered on each pixel -- in other words, the voxel -- before assigning timestamps to the individual events. This paradigm can be found in V2CE, V2E, EventGAN and many other simulators. Our key novelty lies in removing the final, expensive voxel-to-event step. By arguing that the intermediate event stream is unnecessary for many deep learning pipelines, V2V directly generates discrete voxels, fundamentally improving efficiency and avoiding the ill-posed problem of interpolating continuous temporal data. We will add this discussion to our related works section.
>
> **W2. Insuffucient exploration of model design.**
>
> The focus of our paper is to propose a **general** data framework that could boost model performance accross **different tasks** and **different models**. It would indeed be beneficial to explore specific model designs that can better utilize the V2V framework, but this is beyond the scope of our current work.
>
> **W3. Limitations of synthetic datasets.**
>
> Our V2V framework indeed produces synthetic data, and does not provide large-scale real data. However, compared to existing data synthesization methods, the V2V framework scales up the amount of realistic video-based synthetic data, and also allows for more flexible data augmentation methods. This help boosts the performance and robustness of the models trained, which is the source motivation of pursuing large-scale real datasets.
>
> **W4 & Q1. Disentanglement of V2V framework & data scaling.**
>
> The key advantages of our V2V framework are efficiency, scalability and flexibility. The efficiency and scalability contribute to the model performance by allowing the use of large-scale video datasets, so their contributions cannot be disentangled from data scaling. The flexibility, however, can be disentangled and verified.
>
> The flexibility refers to the variety of data augmentation methods applicable when using the V2V framework. For example, in each epoch, a video can be combined with different event thresholds and noise levels, serving as different training samples. Existing event simulators do not support changing thresholds after simulation, so we can evaluate the benefits of the V2V framework by disabling the threshold changing functionality.
>
> Specifically, to disable the threshold changing functionality, we randomly sampled threshold values for each video, and fixed them throughout the training process. We annotate these ablation experiments as "V2V-Fixed" yellow dots in Figure 5. The full methods, with thresholds randomly sampled in each epoch, are annotated as "V2V-Random" green dots.
>
> As visualized in Figure 5 (qualitative results are in Table 2), disabling the threshold changing functionality leads to a performance drop, no matter which scale the training dataset is (100 -> 1K -> 10K). This verifies that the flexibility of the V2V framework contributes to the model performance, disentangled from data scaling.
>
> **Q2. Different performance gains over EVAID and HQF.**
>
> The more significant gains on EVAID compared to HQF stem from EVAID being a more challenging and higher-quality dataset, offering more room for improvement.
>
> 1. EVAID has a higher frame rate (18 ~ 150 FPS) than HQF (17 ~ 26 FPS). This requires models to generalize to higher temporal resolutions and sparser event streams.
>
> 2. EVAID is captured with a Prophesee EKV4 event camera, with resolution 720x1280. HQF is captured with DAVIS346, with resolution 346x260. To gain good performance on both EVAID and HQF, the model would need to generalize to higher spatial resolutions, and also generalize to the different noise characteristics.
>
> 3. There are long periods in EVAID where the camera is static, and no events are generated in the background (an example begins 01:40 of the supplementary video). This requires the model to "remember" the background information reconstructed in the beginning, and distinguish the texture it needs to remember from the noise it should gradually smooth out.
>
> As a result, existing models meet more challenges when tested on EVAID, while the metrics over HQF are near saturation. Hence, our V2V-retrained versions, with better generalization capabilities, can achieve more significant performance gains over EVAID than HQF.
>
> **L1. Missing limitations.**
>
> Due to the space limit, we moved our limitations section to the appendix (Section G). We apologize for the resulting inconvenience.

---

> > ### Comment · Reviewer_bJ8Q · 2025-08-07
> >
> > Thanks for the detailed rebuttal, my concerns have been addressed.

---

> > > ### Author Response · Authors · 2025-08-07
> > >
> > > Thank you for your positive feedback. We will improve our manuscript according to your suggestions.

---

### Official Review · Reviewer_rSE5 · 2025-07-01

**Clarity:** 4
**Significance:** 3
**Originality:** 3
**Rating:** 5
**Confidence:** 5

**Summary:**

The paper introduces Video-to-voxel, a method to generate synthetic event voxels from videos. It builds on the idea that we do not need to calculate actual raw events if we use voxel grids as input to our neural network. The authors show results for two tasks, video reconstruction and optical flow estimation.

**Questions:**

- In Fig. 5/Tab. 1, you are showing a comparison between your method and ESIM-280. How well would a network perform that is trained on WebVid-100, using "conventional" video interpolation + event simulation pipeline? The comparison between V2V + WebVid and ESIM-280 (ECCV'20), seems to limit the comparison.

- What intervals are you choosing for the random contrast thresholds c+, c-?

- Other grid representations have shown performance advantages over voxel grids (e.g., Zubic, ICCV'23). If we neglect the intra-bin time differences for other representations (e.g., time maps), would we see similar effects?

**Ethical Concerns:**

["NO or VERY MINOR ethics concerns only"]

**Final Justification:**

I appreciate the extended experiments on the WebVid-100 dataset. This improves the evaluation. If I see it correctly, models trained on the same data show comparable performance, between video interpolation + vid2e and v2v, while v2v avoids the costly interpolation. I keep my rating at 5.

**Limitations:**

Yes, the authors address limitations.

**Paper Formatting Concerns:**

-

**Quality:**

4

**Strengths And Weaknesses:**

**Strength**
- The idea is simple and tackles an urgent need in event-based vision to enable training on larger-scale datasets.
- The presentation is very clear.
- The results are promising.

**Weaknesses**
- The performance evaluation, compared to a "frame interpolation + event simulation" approach, could be performed a bit cleaner. E.g., use the same base dataset (WebVid-X) and use the best available frame interpolation method. I think the idea is good, but the ESIM-280 might not reflect the performance you could achieve with the "classical" frame interpolation + event simulation approach.

---

> ### Author Rebuttal · Authors · 2025-07-31
>
> **Response to Reviewer rSE5**
>
> Thank you for your positive feedback and insightful suggestions. We will address your concerns below.
>
> **W1 & Q1. Comparison against existing video-based simulators.**
>
> Thank you for your suggestion. In order to compare our V2V framework to a "frame interpolation + event simulation" approach, we conducted two new experiments based on the WebVid-100 dataset.
>
> 1. Frame interpolation: We interpolated the WebVid-100 videos by 8x using the PerVFI [1] model. We selected this model since it was proposed recently (CVPR 2024) and code is available. If required, we can change it to other models.
>
> 	[1] Perception-Oriented Video Frame Interpolation via Asymmetric Blending. Guangyang Wu, Xin Tao, Changlin Li, Wenyi Wang, Xiaohong Liu, and Qingqing Zheng. In Proc. of the IEEE/CVF Conference on Computer Vision and Pattern Recognition (CVPR), 2024.
>
> 2. Event simulation: We used the popular V2E simulator to generate events. We used the "clean" noise mode. For the thresholds, we used the same random thresholds as in the "fixed thresholds" ablation experiment. Since the logarithm mapping of V2E is different from the ESIM simulator (our V2V framework follows ESIM), we scaled the thresholds linearly.
> 	In particular, the logarithm mappings of V2E and ESIM are:
>
> 	```python
> 	def ESIM_log(x):
> 		# The output has range [-6.908, 0], with range length 6.908.
> 		return np.log(0.001 + x / 255.0)
>
> 	def V2E_log(x):
> 		# The output has range [0, 5.545], with range length 5.545.
> 		if x <= 20:
> 			return (1.0 / 20) * math.log(20) * x
> 		else:
> 			return np.log(x)
> 	```
>
> 	So, for a video corresponding to thresholds $c_+$ and $c_-$, we set the V2E thresholds to $c_+ * 5.545 / 6.908$ and $c_- * 5.545 / 6.908$, respectively.
>
> 	In the simulation, a video (1066695484.mp4, with original size 1.2MB) produced 14GB of events. This caused our post-processing step to fail due to out-of-memory issues. Since the rebuttal stage is time-limited, we decided to discard this video, leaving us with a WebVid-99 dataset. The WebVid-99 dataset (in raw events) sums to 12 GB space, while the original WebVid-100 videos sum to 210 MB.
>
> 3. Voxel stacking. We stacked the events into interpolated voxel bins. In this stage, we need to select the durations of the voxel bins. In the V2V experiments, each voxel corresponds to the events between 6 real frames, and the E2VID models reconstruct 24/5 FPS videos. We tried two binning strategies in our V2E experiments:
>
> 	- In strategy 1, we encode all the events between 1+1 original frames (crossing 7 interpolated frames) into a single voxel. This makes the E2VID models reconstruct 24 FPS videos, utilizing the high temporal resolution from frame interpolation. We annotate this as "V2E-1".
>
> 	- In strategy 2, we encode all the events between 5+1 original frames (crossing 35 interpolated frames) into a single voxel. This makes the E2VID models reconstruct 24/5 FPS videos, aligning to the V2V experiments. We annotate this as "V2E-5".
>
> 4. Training and results. During training, we used the same augmentation noise (gaussian noise + hot pixels) as the ESIM protocol. For the V2E-5 experiment, we train for 8000 epochs, aligning to the V2V-WebVid-100 experiment. For the V2E-1 experiment, it produces 5x more training samples, so we train for 1600 epochs.
>
> 	The test results are shown in the table below. The V2V-1 experiment produced better results than the V2V-5 experiment, showing that training on shorter bins may indeed benefit model performance. However, both V2E experiments produced metrics worse than the "WebVid-100-Fixed" experiment, which also uses fixed thresholds for each video.
>
> 	| Idx | Model | Train Dataset    | Loss    | Epochs | HQF MSE | HQF SSIM | HQF LPIPS | EVAID MSE | EVAID SSIM | EVAID LPIPS |
> 	| :-- | :---- | :--------------- | :------ | :----- | :------ | :------- | :-------- | :-------- | :--------- | :---------- |
> 	| (b) | E2VID | WebVid100-Fixed  | V+L1+TH | 8000   | 0.049   | 0.611    | 0.330     | 0.073     | 0.604      | 0.451       |
> 	| (e) | E2VID | WebVid100        | V+L1+TH | 8000   | 0.042   | 0.626    | 0.327     | 0.078     | 0.594      | 0.441       |
> 	| (u) | E2VID | WebVid99-V2E-1   | V+L1+TH | 1600   | 0.057   | 0.524    | 0.459     | 0.116     | 0.484      | 0.561       |
> 	| (v) | E2VID | WebVid99-V2E-5   | V+L1+TH | 8000   | 0.077   | 0.470    | 0.569     | 0.117     | 0.483      | 0.588       |
>
> **Q2. Simulation parameters.**
>
> Details of the selection of simulation parameters are as follows.
>
> 1. We first sample a threshold $c$ from the range [0.05, 2]. Note that we calculate logarithm images with `np.log(0.001 + video/255.0)` (following the ESIM simulator), which means that the amount of change between two frames is in the range of [0, 6.908].
>
> 2. When using real event cameras, it is improbable that the positive and negative thresholds are too different, say 0.1 vs 1. So we uniformly sample the "threshold ratio" $r$ from the range [1, 1.5]. Then, with 50% probability, we set $c_+ = c, c_- = c * r$; with the other 50% probability, we set $c_+ = c * r, c_- = c$. The parameters $c_+$ and $c_- are the positive and negative thresholds, respectively.
>
> 3. The standard deviation of the background gaussian distribution is uniformly sampled from the range [0, 0.1].
>
> 4. The standard deviation of the hot pixel gaussian distribution is uniformly sampled from the range [0, 10].
>
> 5. The fraction of hot pixels is uniformly sampled from the range [0, 0.001] (i.e., 0 to 0.1%).
>
> These configuration details will be included in our code release to insure reproducibility.
>
> **Q3. Other grid representations.**
>
> The idea of using the V2V framework on other representations by neglecting intra-bin differences is interesting. Intuitively, we believe that this could work. However, further experiments would be needed for validation.

---

> > ### Comment · Reviewer_rSE5 · 2025-08-05
> >
> > I appreciate the extended experiments on the WebVid-100 dataset. This improves the evaluation. If I see it correctly, models trained on the same data show comparable performance, between video interpolation + vid2e and v2v, while v2v avoids the costly interpolation. I keep my rating at 5.

---

> > > ### Author Response · Authors · 2025-08-06
> > >
> > > Thank you for your valuable time and positive feedback. We are glad that the additional experiments were helpful for the evaluation. Thanks again for your review!

---

### Official Review · Reviewer_92jc · 2025-07-03

**Clarity:** 4
**Significance:** 4
**Originality:** 3
**Rating:** 5
**Confidence:** 4

**Summary:**

This paper proposes a method to generate event-based training data from conventional RGB video. The method directly converts RGB images into an event voxel representation, skipping the step of materializing sparse event data. This significantly reduces the storage and memory requirements, as event voxels are a much more compact representation than sparse events. Using this method, the authors generate large-scale event-based training datasets. Training on these larger-scale datasets leads to substantial improvements in model performance.

**Questions:**

My only request is further explanation/justification for the modeling assumptions in Equation 5. I am basing my positive rating on the assumption that this model is generally sound. I am interested to see the comments of the other reviewers on this point.

**Ethical Concerns:**

["NO or VERY MINOR ethics concerns only"]

**Final Justification:**

I remain positive on this paper - the authors have addressed my few concerns in the rebuttal.

**Limitations:**

The paper is missing a limitations section. A potential discussion point would be issues arising from using lower-speed RGB video to generate training data. E.g., have the authors observed any generalization failures on real event data? Are there types of motion that are not well represented in the generated data?

**Paper Formatting Concerns:**

No major concerns

The parentheses and floor symbols in Equation 6 should be full-height - \left(, \right), \lfloor, and \rfloor.

**Quality:**

3

**Strengths And Weaknesses:**

**Strengths**

I am overall positive on this paper. The authors have identified one of the key issues with event cameras - smaller-scale datasets hinder model performance, which inhibits further adoption of event cameras. The proposed method can be used to train event camera models on internet-scale data, and hopefully bring event cameras into the realm of modern large-scale deep learning. The results are convincing (especially the qualitative results in Figure 6).

The comparison between interpolated and discrete event voxels was nicely explained and well-illustrated in the figure.

The writing and figure quality is excellent.

I appreciate the transparency about the experiment mistake at L287.

**Weaknesses**

The description of key technical details (in the last half of page 6) could be expanded. In particular, I would like to see more discussion of Equation 5 and justification for its modeling assumptions. E.g., why can we assume that background noise can be modeled as a Gaussian on the output voltage?

The temporal resolution of the event voxels is tied to the input frame rate. This could be seen either as a feature or a bug - the method is not capable of producing higher-speed outputs, but we also don't have to solve the interpolation problem.

---

> ### Author Rebuttal · Authors · 2025-07-31
>
> **Response to Reviewer 92jc**
>
> Thank you for your positive evaluation and constructive feedback. We will address your concerns below.
>
> **S4. Experiment mistake.**
>
> Thank you for your understanding. We have reconducted the experiment with the correct configuration, and we will update the results in the final version of the paper. The test results before and after correction are as follows.
>
> | Idx       | Model | Train Dataset | Loss   | Eps | HQF MSE | HQF SSIM | HQF LPIPS | EVAID MSE | EVAID SSIM | EVAID LPIPS |
> | :-------- | :---- | :------------ | :----- | :-- | :------ | :------- | :-------- | :-------- | :--------- | :---------- |
> | (k-old)   | ETNet | WebVid10K*    | V+L1+H | 100 | 0.036   | 0.662    | 0.279     | 0.081     | 0.586      | 0.432       |
> | (k-fixed) | ETNet | WebVid10K     | V+L1+H | 100 | 0.039   | 0.641    | 0.306     | 0.056     | 0.610      | 0.406       |
>
> **W1 & Q. Justification for noise simulation modeling.**
>
> Our noise model was designed by referencing two widely used simulators, ESIM and DVS-Voltmeter. The Gaussian noise added to the output voltage is adapted from DVS-Voltmeter's model, which simulates the voltage change as a Brownian motion process caused by photon reception randomness. This is more physically plausible than adding noise post-voxelization (as in the ESIM training protocol), as it allows noise to interact with the signal during event generation. For simplicity, we did not include other noise types like leak noise, but they could be easily incorporated into our framework.
>
> **W2 & L1. Adaption to higher temporal resolution.**
>
> When applying V2V-trained models to real event streams, the temporal resolution of the outputs can be adjusted by changing the bin durations of the voxels. For example, the EVAID dataset has ground truth frame rates up to 150 FPS (see Table 5), so we set the duration of each bin to 1/150/5 = 1.33 ms. The resulting high frame rate video outputs achieved good results qualitatively (Figure 6) and quantitatively (Table 2).
>
> This raises a question: Why can the model, trained on 5 FPS synthetic voxels, adapt to 150 FPS real event inference? We believe the answer lies in the vast diversity of our training data. The WebVid dataset contains scenes with a wide range of motion speeds, from rapid action to nearly static shots. Statistically, voxels from long bins in slow scenes can appear similar to voxels from short bins in fast scenes. This diversity allows the model to learn representations that are robust to different bin durations and, therefore, different temporal resolutions.
>
> **L1. Missing limitations.**
>
> Due to the space limit, we moved our limitations section to the appendix (Section G). We apologize for the resulting inconvenience.

---

> > ### Comment · Reviewer_92jc · 2025-08-07
> > **Response to Rebuttal**
> >
> > Thanks to the reviewers for their response - and my apologies for missing the limitations section in the appendix.
> >
> > My primary concern (albeit minor) was justification for the modeling assumptions, and this has been addressed in the rebuttal. It would be great to include some version of this paragraph in the final manuscript.

---

> > > ### Author Response · Authors · 2025-08-07
> > >
> > > Thank you for your positive feedback and helpful suggestion. We will add more details about the modeling in our revised manuscript to improve clarity.

---

> ### Comment · Area_Chair_weaw · 2025-08-06
>
> Dear 92jc, I see your review is positive, but we need you to show up and say something that engages with the rebuttal.

---

### Official Review · Reviewer_WNjQ · 2025-07-08

**Clarity:** 2
**Significance:** 3
**Originality:** 3
**Rating:** 4
**Confidence:** 4

**Summary:**

Scaling up event-based training dataset is tough because of massive bandwidth/storage requirements of synthetic data pipelines. The authors propose a Video2Voxel approach which converts video frames to event-based voxel grids – bypassing event stream generation.
Event cameras emit an asynchronous stream of events $(x, y, t, p)$, where $(x, y)$ are pixel indices, t is a microsecond-accurate timestamp, and $p \in {+1, –1}$ indicates whether the log-intensity at that pixel increased or decreased beyond a preset threshold. Pixels fire only when that contrast change occurs, so the data are spatially sparse and temporally precise. Frame cameras, by contrast, deliver full RGB / grayscale images at discrete, globally clocked intervals (e.g. 30 fps). Peak bandwidth in an event stream can be high under rapid motion, yet average bandwidth is typically lower than that of a high-speed conventional camera because inactive pixels remain silent. One can also view it as a temporal-gradient sensor [a]: integrating events over time tells us brightness, but only up to an unknown offset at each pixel.

Earlier methods wish to use videos either via (a) video simulation which uses frame interpolation to detect events, this is of limited fidelity because it’s an attempt to fill event reconstruction gaps from data that is not present in low-temporal-resolution videos (b) model simulation, i.e. build a simulation environment in 3D and render event camera streams. While viable, this is tough to use because of the sim2real gap. When synthetic streams are persisted at micro-second resolution the I/O load explodes, whereas on-the-fly processing of real sensors is typically lightweight.

The paper asks the simple question: “Do I need event streams in the first place?” Many current event-based methods convert async-event into dense representations like voxel grids (essentially an accumulation step for each sparse observation $(x, y, t, p)$). Thus, instead of generating event streams, one can generate voxel accumulations. This has several advantages (a) one can now use all the video data without interpolation (b) it’s more efficient to generate compared to streams, which are anyways accumulated later.
Paper focuses on dense prediction tasks: event-based video reconstruction and optical flow estimation. The method is to utilize the difference between interpolated and discrete voxel grids – recognizing that interpolated grids generated from videos do not match event cameras compared to discrete voxel grids. Using this key fact, they write a simulator that provides them discrete voxel grids from videos.

[a] Why I want a Gradient Camera, CVPR 2005

**Questions:**

While V2V convincingly scales training data for voxel-based reconstruction and optical-flow networks, it does not show benefits for any system that leverages an event camera’s fundamental advantage—microsecond-level latency. Tasks that need per-event timing (event-based SLAM, gesture recognition, etc.) remain completely untested. The paper therefore answers an important data-volume question for a subset of event-vision problems, but it sidesteps—and leaves open—whether its video-to-voxel shortcut actually helps the applications for which event cameras were invented.

In principle the approach could help, say, by augmenting SLAM with the authors’ flow estimator, but we need to see such a system-level demonstration before the claim is convincing. Evaluating only on dense voxel tasks tells us little if their methods leverages the advantages of both internet-scale data and event cameras for the things they are uniquely good at. Can the authors show improvements on at least one latency-critical, event-native task?

I'll switch to an accept if such evidence is provided. The core idea is in itself solid.

**Ethical Concerns:**

["NO or VERY MINOR ethics concerns only"]

**Final Justification:**

Authors have resolved many of my concerns by providing additional clarifications about existing results and have promised to improve their presentation -- which amounts to a few minor changes. They also provided additional results by testing their synthetic data pipeline in comparison to training on limited real data from scratch and shown their large-scale data-driven method prevails. Lastly, while the additional pipeline experiment is limited in scope and size, it provides some hope that large-scale data-driven priors from videos can be exploited for fast paced event camera pipelines.

**Limitations:**

Yes

**Quality:**

3

**Strengths And Weaknesses:**

1. Strength. Motivation, problem statement, and method are all sound. It makes sense to directly generate voxels if methods resort to voxels to accumulate events in the first place when operating on event streams.

2. Strength. Results themselves appear to be sound for reconstruction and optical flow estimations. Efficiency gains are reported and are appreciated. In itself, for these two tasks, the quantitative gains are solid.

3. Weakness. Paper writing leaves much to be desired. Reframe inherent ambiguity in event-based vision tasks. Please cite [a] or other papers to reference the inherent ambiguity, this is a pretty well known fact about event cameras.

4.	Weakness. Results section is hard to read, and parse. It reads like a technical report, and the onus is on the reader to understand what are the distinct advantages of their approach. It can be re-written to highlight the benefits of their video2event-voxel method.

- Table 2 to is extremely hard to parse, which results are yours and which ones are the baselines. I understand WebVid10K is yours, but reorganize the table to highlight the results and other methodologies (for example ESIM-280-Intp. Seems to be a video interpolation method).
- Fig 6, Please highlight which methods are yours and which ones are others in the figure/caption.

5. Weakness. Evaluation scope is narrow. So, the method is simple and improves on dense tasks, but what’s missing from the overall picture? Voxel accumulation helps in tasks where an explicit accumulation step is required in the first place – this is limited as any dense binning costs temporal precision but keeps the full spatial grid, which defeats the entire purpose of an event camera in my view. Essentially, the whole point of the event camera was to gain temporal resolution by sacrificing per-frame intensity, and performing an accumulation step is counter to the real benefit of this camera.
- The real core solution in my view addresses the following question -- can we reuse weights trained on such video scale datasets, and finetune on limited real events data for other event tasks (on, say some sparse task or some system level program which operates on events, say event based SLAM), and get benefits over the limited data regime?

---

> ### Author Rebuttal · Authors · 2025-07-31
>
> **Response to Reviewer WNjQ**
>
> Thank you for your thorough review and insightful comments. We will address each concern below.
>
> **W3 & W4. Clarity of writing.**
>
> We will improve the clarity of our writing in the final version.
>
> * We will add citations (including [a]) to our preliminary discussion on the inherent ambiguity of event-based vision, as suggested.
> * We will highlight the "Ours" methods in Table 2 and Figure 6 to improve readability.
>
> The "ESIM-280-Intp" in Table 2 refers to a training dataset created from ESIM, composed of 280 scenes, and converted to the Interpolated voxel representation (L259-L261). The corresponding experiments (l) and (m) are also video reconstruction experiments. They are designed to compare the interpolated voxel representation against the discrete voxel representation. We apologize for any confusions.
>
> **W5 & Q. Limitations of the voxel representation.**
>
> We acknowledge the reviewer's concern on discarding intra-bin timing, which is a limitation we discuss in Section G.
> However, we believe that this limitation does not "defeat the entire purpose of an event camera".
>
> 1. Voxel representations preserve key advantages of event cameras. As detailed in Section 3.2, while discrete voxels discard fine-grained intra-bin timing, they preserve essential inter-bin temporal dynamics. This allows for:
>
> 	* High temporal resolution: By adjusting bin durations, voxel grids can effectively encode high-speed motion. In our work, we successfully reconstruct videos at 150 FPS on the EVAID dataset.
> 	* High dynamic range (HDR): The HDR property of event cameras, which relies on per-pixel logarithmic changes, is fully preserved in the voxel representation.
> 	* Sparsity and efficiency: The principle of sparsity can be maintained by adapting bin durations based on event activity or by processing only local patches with sufficient event density, preserving the low-power potential of event cameras.
>
> 2.  Voxel-based methods are widely and successfully used across diverse event-based tasks, including latency-sensitive ones like SLAM. The simplicity and effectiveness of voxel grids have made them a cornerstone of modern event-based vision research. Examples include:
>
> 	* SLAM: [CVPR 2019] Unsupervised Event-based Learning of Optical Flow, Depth, and Egomotion.
> 	* Action recognition: [CVPR 2022] E^2(GO)MOTION: Motion Augmented Event Stream for Egocentric Action Recognition.
> 	* Head pose estimation: [ECCV 2024] Event-based Head Pose Estimation: Benchmark and Method.
> 	* Object detection: [CVPR 2024] Scene Adaptive Sparse Transformer for Event-based Object Detection.
> 	* Eye tracking: [CVPRW 2024] Event-Based Eye Tracking. AIS 2024 Challenge Survey.
> 	* HDR imaging: [CVPR 2023] Learning event guided high dynamic range video reconstruction.
> 	* Image deblurring: [CVPRW 2025] NTIRE 2025 Challenge on Event-Based Image Deblurring: Methods and Results.
>
> 	Our V2V framework can boost models that accepts a voxel-based input, making it broadly applicable. We chose video reconstruction and optical flow as representative examples, and we hope to inspire the community to explore more tasks given task-specific large-scale datasets.
>
> 3. While the ultimate goal may be end-to-end spiking neural networks on neuromorphic hardware, the current ecosystem faces challenges with hardware cost and computational limitations. High-performing, GPU-trained voxel-based models can demonstrate the value of event cameras, driving the widespread adoption needed to scale the industry. As the hardware matures, these successes can pave the way for a transition to more neuromorphic approaches.
>
> 4. Regarding whether training on synthetic data can improve performance on real data, our intuition is that it can. Existing literature, such as *[CVPR 2020] Video to Events: Recycling Video Datasets for Event Cameras*, suggest that training on synthetic data and fine-tuning on real data often yields the best results. However, this is beyond the scope of our current study and awaits future validation.

---

> ### Comment · Reviewer_WNjQ · 2025-08-01
> **Results on one transferable task or some end-to-end system using event data?**
>
> Thank you for the response!! Let me first describe my rationale for the rating despite strong results and good presentation -- The contribution of the paper is a system-level insight, i.e. "instead of generating event streams, one can generate voxel accumulations, and this allows us to leverage videos while not blowing up I/O and other costs during training". Closing the loop, i.e. an end-to-end evaluation actually tells us if the model trained on all of these videos is actually useful in real situations or not.
>
> On limitations of voxel representations -
> (a) Agreed on Point 1 -- real event data does preserve inter-bin temporal dynamics. Your training simulates event data from videos with lower temporal resolution as far as I understood the paper. Are you claiming when you "successfully reconstruct videos at 150 FPS on the EVAID dataset", it's done by training on only 30 FPS videos? This does partially resolve my concern if it's the case.
>
> (b) Yes, voxel based methods are used in prior work for downstream tasks -- agreed. I think my core issue is that it would be much more impactful to show that your method enables leveraging thousands and millions of videos on one of these tasks (or improved generalization to a different dataset for the same task).
>
> (c) Agreed.
>
> (d) I disagree this is beyond the scope of the problem. The whole point of the work is to leverage all of these thousands (and potentially millions of) videos -- which result in synthetic data is to help event camera related tasks. As pointed in [CVPR 2020, Gehrig et al] which shows corroborating experiments, I'm asking the same question "can we reuse weights trained on such video scale datasets, and finetune on limited real events data for other event tasks". Experiments along the line of [Gehrig et al] would significantly improve the work.
>
> Feel free to correct me if I have missed something about this work and in my assessment.

---

> > ### Author Response · Authors · 2025-08-02
> >
> > Thank you for your careful considerations. We will further discuss the points below.
> >
> > **(a) Clarification on temporal resolution.**
> >
> > Actually, our models are trained on ~ 5 FPS synthetic data: the videos are 24 FPS, and we take the interval between 2 frames as one voxel bin, so a full voxel (with 5 bins) covers a time span over 5/24 = 0.21 seconds. We tested it on 150 FPS EVAID sequences: each full voxel covers 1/150 seconds, which means each bin covers 1/(150*5) seconds. This creates a **30x** temporal resolution gap between training and testing, but the models generalize well.
> >
> > We believe the ability to generalize sources from the vast diversity of our training data. The WebVid dataset contains scenes with a wide range of motion speeds, from rapid action to nearly static shots. Statistically, voxels from long bins in slow scenes can appear similar to voxels from short bins in fast scenes. Hence, the scene diversity can help the model learn to generalize across different temporal resolutions.
> >
> > **(b) Leveraging large-scale video datasets.**
> >
> > We apologize for any misunderstanding. Our intention is not to use the same WebVid video dataset for all downstream tasks, but rather to leverage the V2V framework in combination with other domain-specific large-scale video datasets. This would allows us to utilize the rich annotation resources available in the video domain.
> >
> > For example, for event-based object detection, the most commonly used real datasets are 1Mpx [1], Gen1 [2] and DSEC-Det [3]. Although 1Mpx and Gen1 are large in scale, their diversity is limited: they are focused on driving scenarios, and only 2-3 categories (pedestrians, cars, two-wheel vehicles) are annotated. (We could not find the number of sequences of these datasets.)
> >
> > In the video domain, however, there are large-scale and diverse segmentation datasets. (Note that we can acquire detection labels from segmentation datasets, since segmentation masks are more informative than bounding boxes.) For example, the BURST [5] and LV-VIS [6] datasets are diverse datasets covering a wide range of categories, aiming for open-vocabulary video segmentation. The SA-V dataset [7] provides segmentation masks over even more diverse sequences, although no text categories are provided.
> >
> > The comparison between the datasets are as follows. By utilizing the V2V framework, we would be able to leverage the vast diversity of these video datasets, pushing the application of event-based object detection to new domains.
> >
> > | Dataset | Ev/Vid | Source | Categories | Sequences | Duration | # Masks | # B-Boxes |
> > | --- | --- | --- | --- | --- | --- | --- | --- |
> > | 1Mpx [1] | Ev | NeurIPS 2020 | 3 | ? | 14.7 hr | / | 25M |
> > | Gen1 [2] | Ev | arXiv 2020 | 2 | ? | 39 hr | / | 255k |
> > | DSEC-Det [3] | Ev | RA-L 2022 | 8 | 60 | 1 hr | / | 390k |
> > | WOD [4] | Vid | CVPR 2020 | 4 | 1150 | 6.4 hr | / | 9.9M |
> > | BURST [5] | Vid | WACV 2023 | 482 | 2914 | 24.3 hr | 600k | / |
> > | LV-VIS [6] | Vid | ICCV 2023 | 1196 | 4828 | 6.2 hr | 544k | / |
> > | SA-V [7] | Vid | ICLR 2025 | None | 50.9k | 196.0 hr | 35.5M | / |
> >
> >
> > [1] Etienne Perot, Pierre De Tournemire, Davide Nitti, Jonathan Masci, and Amos Sironi. Learning to detect objects with a 1 megapixel event camera. Advances in Neural Information Processing Systems (NeurIPS), 2020.
> >
> > [2] Pierre De Tournemire, Davide Nitti, Etienne Perot, Davide Migliore, and Amos Sironi. A large scale event-based detection dataset for automotive. arXiv preprint arXiv:2001.08499, 2020.
> >
> > [3] Daniel Gehrig and Davide Scaramuzza. Low Latency Automotive Vision with Event Cameras. Nature, 2024.
> >
> > [4] Pei Sun, Henrik Kretzschmar, Xerxes Dotiwalla, Aurelien Chouard, Vijaysai Patnaik, Paul Tsui, et al. Scalability in Perception for Autonomous Driving: Waymo Open Dataset. Conference on Computer Vision and Pattern Recognition (CVPR), 2020.
> >
> > [5] Ali Athar, Jonathon Luiten, Paul Voigtlaender, Tarasha Khurana, Achal Dave, Bastian Leibe, and Deva Ramanan. BURST: A Benchmark for Unifying Object Recognition, Segmentation and Tracking in Video. Winter Conference on Applications of Computer Vision (WACV), 2023.
> >
> > [6] Haochen Wang, Cilin Yan, Shuai Wang, Xiaolong Jiang, Xu Tang, Yao Hu, Weidi Xie, and Efstratios Gavves. Towards Open-Vocabulary Video Instance Segmentation. International Conference on Computer Vision (ICCV), 2023.
> >
> > [7] Nikhila Ravi, Valentin Gabeur, Yuan-Ting Hu, Ronghang Hu, Chaitanya Ryali, Tengyu Ma, et al. SAM 2: Segment Anything in Images and Videos. International Conference on Learning Representations (ICLR), 2025.

---

> > > ### Author Response · Authors · 2025-08-02
> > >
> > > **(d) Reusing weights by finetuning on real events.**
> > >
> > > We think that more detailed discussion is needed to answer this question. Let A and B be two different tasks (such as video reconstruction and object detection), A_V and B_V annotate their corresponding video datasets (for synthetic voxel generation), and A_1, A_2, B_1, B_2 annotate real event datasets for these tasks.
> > >
> > > - What we proved feasible in this paper:
> > >
> > >   - By training a model on A_V for task A, it can generalize to real datasets A_1 and A_2.
> > >
> > >   - To utilize the V2V framework on task B, we can train another model on B_V, and make it work on B_1 and B_2.
> > >
> > > - What we can validate about finetuning:
> > >
> > >   - If we further finetune the A_V-trained model on a train split of A_1, then it could work better on a test split of A_1.
> > >
> > >   - However, we fear that finetuning on A_1 will not lead to performance gains on A_2. A_V has a large diversity, with different noise models and thresholds included. If we finetune it on A_1, which has a specific camera model, it may learn to specialize on it and lose its generalization ability to A_2.
> > >
> > >   - If required, we can conduct an experiment by finetuning V2V-E2VID on a split of EVAID (HQF), and show performance gains on the rest of EVAID (HQF).
> > >
> > > - How we can reuse weights on other tasks:
> > >
> > >   - These trained models can be used as components for other task pipelines. For example, to conduct event-based object detection, we can use an E2VID model to convert events to images, then use off-the-shelf image-based object detection models.
> > >
> > >   - If required, we can conduct an object detection experiment, showing that using a better E2VID model (trained with the V2V framework) leads to a better event-based object detection pipeline.
> > >
> > > - What is beyond the scope of this research:
> > >
> > >   - We do not know how to finetune a model trained on A_V, for task A, to work for task B. Different tasks have different network structures, and the parameters are not directly transferable.
> > >
> > >   - Although the idea of cross-task transfer learning is promising, it is still an open problem, and we do not expect to solve it within this paper.

---

> ### Comment · Reviewer_WNjQ · 2025-08-02
> **Thanks for the response!**
>
> (a) Thanks for this critical point, maybe I missed it from the paper but it does help my understanding a lot. This is good to know to that the model generalizes to temporal resolutions beyond what it's trained for.
>
> (b) Yes, this makes sense. I think my question is better answered in response to (d), where we discuss the evaluation.
>
> (d) Discussion on evaluation.
>
> (i) Okay on re-reading the paper, I see that your evaluation on A_1 (HQF) and A_2 (EVAID) (Table 2) is in some sense zero-shot, not trained on the real dataset at all -- very cool. ESIM is the event data simulator, whose performance is worse. Can the authors clarify why ESIM is tagged with "Pretrain" -- is the model then trained on HQF or EVAID? Please correct me if I'm wrong in my understanding. In general, Table 2 is very hard to interpret because there is experimental combinations/ablations that are all shown together, it's difficult to understand the effect of one specific axis. Breaking the table into a few tables (some for the data size ablations or loss ablations) would help bring clarity and surface up these points better.
>
> (ii) Worry of overfitting on a specific dataset is reasonable -- I'm not hoping that the finetuned model would generalize to both the datasets. It would be nice to see if fine-tuning on a specific dataset additionally helps, and it's an insight that strengthens the argument for (b) - Training on WebVid + Small Event Dataset >> Training on Small Event Dataset.
>
> (iii) This is the kind of evaluation/validation I'm hoping for -- to know for a fact that a model trained on diverse even if synthetic data helps in a better event-based object detection (or some other method, like SLAM) pipeline. Ideally the pipeline ought to be event "native" -- uses event streams by accumulating them and also in other ways, but I suppose most pipelines resort to accumulation (point 2 in author's initial response).
>
> (iv) Noted. I do think a general transformer like model is a ripe direction in that case to simplify parameter sharing across tasks.

---

> > ### Author Response · Authors · 2025-08-03
> >
> > Thank you for your detailed discussion. We will further clarify the points below.
> >
> > **(i) Meaning of "Pretrain".**
> >
> > We apologize for the confusion. All of our experiments are zero-shot, and "Pretrain" only refers to the source of the model weights.
> >
> > For the baseline methods, we directly downloaded the pretrained weights provided by the E2VID/HyperE2VID/ETNet papers, reproducing their test results. According to the original papers, these models were only trained on the ESIM-280 dataset. Hence we annotate them as "Pretrain-ESIM".
> >
> > To conduct ablation experiments, such as experiments (l) and (m), we trained our own models from scratch on the ESIM-280 dataset.
> >
> > We felt the need to distinguish these two cases, but we did not make the notation clear enough. We will clarify this in the final version of the paper.
> >
> > Thank you for your suggestion on breaking the table. In our revised version, we will separate the experiments in our table to improve clarity.
> >
> > **(ii) Finetuning on a specific dataset.**
> >
> > Thank you for your understanding. We will conduct the experiment in the following days.
> >
> > **(iii) Reusing weights on other tasks.**
> >
> > Thank you for your understanding. We will conduct the experiment in the following days.

---

> > ### Author Response · Authors · 2025-08-06
> > **Experiment results**
> >
> > Thank you for your patience. The experiments have been conducted, and the details are as follows.
> >
> > **(1) Finetuning E2VID on EVAID.**
> >
> > Our EVAID test set has 10 sequences. We took 7 sequences (Ball, Box, Building, Outdoor, Playball, Room1, Toy, Wall) as a train set, and 3 sequences (Bear, Sculpture, Traffic) as the new test set.
> >
> > 1. Zero-shot inference:
> >
> >     We directly used the V2V-E2VID model weights, trained on the WebVid dataset, for zero-shot inference on the EVAID test set.
> >
> > 2. Finetuning on EVAID:
> >
> >     We finetuned the pretrained V2V-E2VID model on the EVAID train set for 100 epochs, with a learning rate of 0.0001, observing that the validation loss had converged.
> >
> > 3. Directly training on EVAID:
> >
> >     We trained an E2VID model from scratch on the EVAID train set. We trained it for 2000 epochs with learning rate 0.0001, observing that the validation loss had converged.
> >
> > The resulting metrics of the three models are as follows:
> >
> > | Model | MSE (↓) | SSIM (↑) | LPIPS (↓)|
> > | --- | --- | --- | --- |
> > | V2V + Zero-shot | **0.056** | **0.661** | **0.424** |
> > | V2V + Finetune | 0.064 | 0.645 | 0.458 |
> > | Trained on real | 0.078 | 0.483 | 0.658 |
> >
> > Surprisingly, the zero-shot model performed best. Finetuning caused a slight performance drop, while training on real data from scratch performed worst. This is likely due to the small size of the EVAID dataset, which caused the model to overfit to the training sequences and fail to generalize to the test sequences.
> >
> > **(2) Applying weights to downstream tasks.**
> >
> > In order to demonstrate the reusability of the V2V weights on downstream tasks, we conducted an experiment on the N-MNIST classification task.
> >
> > The MNIST dataset is an image-based dataset, corresponding to the task of classifying hand-written digits, frequently used as a toy dataset in computer vision. N-MNIST [1] is a corresponding event-based classification dataset, recorded by saccading the original MNIST images with an real event camera. We chose this task due to its convenience: the dataset is small and code is simple, so we could conduct the experiment as quickly as possible.
> >
> > We first trained a very simple convolution model on the train split of MNIST (the image version). On the test split (images), the model achieved an accuarcy of **98.76%**.
> >
> > Then we explored using video reconstruction models in a zero-shot pipeline for the N-MNIST task. The pipeline is as follows:
> >
> > 1. Take real event streams from the N-MNIST test split. Stack them to 5*5 event bins.
> >
> > 2. For each of the 5 voxels (each with 5 bins), use the E2VID model to reconstruct an image frame. This produces 5 reconstructed frames.
> >
> > 3. Use the last frame as the input to the image-based MNIST classifier, and predict the digit.
> >
> > When using the original E2VID model, the resulting accuracy on the N-MNIST test set is **47.46%**. When changing to the V2V-E2VID model, the accuracy improves to **63.62%**. This is due to the better reconstruction quality of the V2V-E2VID model.
> >
> > This shows that the quality improvement brought by the V2V framework can be utilized by zero-shot downstream tasks. Note that there are existing classification models that can generalize much better; the purpose of using a weak model is to compare the original E2VID against V2V-E2VID.
> >
> > [1] Orchard, G.; Cohen, G.; Jayawant, A.; and Thakor, N.  “Converting Static Image Datasets to Spiking Neuromorphic Datasets Using Saccades", Frontiers in Neuroscience, vol.9, no.437, Oct. 2015.

---

> > > ### Comment · Reviewer_WNjQ · 2025-08-06
> > > **Looks good!**
> > >
> > > I believe most of my concerns are resolved to a reasonable extent. I do hope authors note the limitations of these experiments in their final manuscript apart from the additional results. I'll update my rating to positive.

---

> > > > ### Author Response · Authors · 2025-08-06
> > > >
> > > > Thank you for your valuable feedback and support. We will revise the manuscript as you suggested, specifically by clarifying the limitations of our work and reorganizing the results section.

---

### Note · Authors · 2025-08-13

We sincerely thank the reviewers and ACs for their time and constructive feedback throughout the review process. We are encouraged that reviewers broadly recognized our work's strengths: the "motivation, problem statement, and method are all sound" (Reviewer WNjQ), the "writing and figure quality is excellent" (Reviewer 92jc), and that our approach "tackles an urgent need in event-based vision to enable training on larger-scale datasets" (Reviewer rSE5).

The author-reviewer discussion was highly productive. We addressed key concerns by providing clarifications and experiment results, which were well-received:

* To address Reviewer WNjQ's concerns about evaluation scope, we provided experiment results on downstream tasks and generalization. Our results demonstrated that: (1) zero-shot V2V models can outperform models finetuned on small-scale real data, proving V2V's superior generalization capabilities through vast data diversity; and (2) the V2V framework has broad applicability to other tasks such as object detection, with this generalization ability improving the performance of downstream tasks like classification. The reviewer found this evidence compelling, stating, "most of my concerns are resolved... I'll update my rating to positive."

* In response to Reviewer rSE5's suggestion, we provided direct experimental comparison results against a traditional interpolation-based pipeline, which the reviewer confirmed "improves the evaluation."

* We also successfully addressed the remaining concerns from Reviewer bJ8Q (on novelty and experimental disentanglement) and Reviewer 92jc (on modeling assumptions), with both reviewers confirming that their concerns "have been addressed."

In light of the discussion, we commit to the following improvements in the final manuscript:

* We will add the experiment results provided during the rebuttal, including the downstream task evaluation and the direct comparison with the V2E-based pipeline.

* We will reorganize the results section (especially Table 2) and add justifications for our noise model, as suggested.

* We will add the requested citations, fix minor formatting, and ensure the limitations are discussed in the main paper.

We believe that our V2V framework can boost efficiency and performance in a large variety of event-based vision tasks, and we hope that it will help the community advance towards the goal of effective and efficient neuromorphic vision.

---

### Decision · Program_Chairs · 2025-09-17

**Decision:**

Accept (poster)

**Comment:**

This paper proposes a method to generate event-based training data from conventional RGB video, by converting a video into (synthetic) event voxel grids. The paper received a consensus of positive reviews (4, 5, 5, 4). The reviewers appreciate the main idea of the work, and appear excited by the possibility of training event camera models on internet-scale data. While they note that the current set of experiments is still limited in scope and size, they find that the experiments added in the rebuttal improved their impression of the work's value, and everyone's overall opinion is positive. The authors also promised to make minor revisions to the manuscript, clarifying the limitations and reorganizing the results for clarity. Based on all this, the AC recommends acceptance.